

# Future changes in the extratropical storm tracks and cyclone intensity, wind speed, and structure

Matthew D. K. Priestley[1] and Jennifer L. Catto[1]

[1]College of Engineering, Mathematics and Physical Sciences, University of Exeter, Exeter, UK

**Correspondence:** Matthew D. K. Priestley (m.priestley@exeter.ac.uk)

**Abstract.** Future changes in extratropical cyclones and the associated storm tracks are uncertain. Using the new CMIP6 models, we investigate changes to seasonal mean storm tracks and composite wind speeds at different levels of the troposphere for the winter and summer seasons in both the Northern (NH) and Southern Hemispheres (SH). Changes are assessed across four different climate scenarios. The seasonal mean storm tracks are predicted to shift polewards in the SH and also in the North Pacific, with an extension into Europe for the North Atlantic storm track. Overall, the number of cyclones will decrease by ~5% by the end of the 21st century, although the number of extreme cyclones will increase by 4% in NH winter. Cyclone wind speeds are projected to strengthen throughout the troposphere in the winter seasons and also summer in the SH, with a weakening projected in NH summer, although there are minimal changes in the maximum wind speed in the lower troposphere. Large amounts of this change can be associated with changes in the speed of cyclones in the future. Changes in wind speeds are concentrated in the warm sector of cyclones and the area of extreme winds may be up to 40% larger by the end of the century. The largest changes are seen for the SSP5-85 scenario, although large amount of change can be mitigated by restricting warming to that seen in the SSP1-26 and 2-45 scenarios. Extreme cyclones show larger increases in wind speed and peak vorticity than the average strength cyclones, with the extreme cyclones showing a larger increase in wind speed in the warm sector.

## 1 Introduction

Extratropical cyclones are the key driver of day-to-day weather variability in the mid-latitudes and can be associated with significant impacts from extreme winds (Browning, 2004) and precipitation (Hawcroft et al., 2012). In the most recent assessment report from the IPCC (AR5 Christensen et al., 2013) it was stated that it is likely there will be a poleward shift of storm tracks in the Southern Hemisphere (SH) and North Pacific, however, there is low confidence in the magnitude of regional storm track changes. With the recent availability of CMIP6 models (Eyring et al., 2016) a renewed assessment of these changes can be made.

Changes in the seasonal mean storm tracks are relatively robust and have been consistent across a number of generations of GCMs (Yin, 2005; Chang et al., 2012, 2013; Colle et al., 2013; Zappa et al., 2013; Harvey et al., 2020). In the SH a robust poleward shift of the storm track is projected (Bengtsson et al., 2006; Chang et al., 2012; Chang, 2017), with the pattern of change being less clear in the Northern Hemisphere (NH). In the NH there is a projected shift of the storm track poleward in the western North Pacific and an extension of the North Atlantic storm track into Europe (Bengtsson et al., 2006; Zappa





et al., 2013; Harvey et al., 2020). Furthermore, a reduction in the number of cyclones over the Mediterranean is projected (Zappa et al., 2015). The differing responses in the NH and SH are largely a result of opposing forcings in the upper and lower troposphere in the NH (Shaw et al., 2016; Shaw, 2019) and the role of low-level, high latitude heating in shifting the storm tracks equatorwards (Butler et al., 2010). Using the Eulerian definition of the storm tracks, CMIP6 models have shown to have
larger projected changes than CMIP5 (Harvey et al., 2020). These larger changes are thought to be a result of the higher climate sensitivity of CMIP6 models, relative to CMIP5 (Zelinka et al., 2020). Currently, no Lagrangian perspective on storm track changes in CMIP6 models has been performed.

It is widely agreed that the total number of extratropical cyclones will decrease in the future (Geng and Sugi, 2003; Bengtsson
et al., 2006; Catto et al., 2011; Mizuta et al., 2011; Chang et al., 2012; Zappa et al., 2013; Michaelis et al., 2017; Sinclair et al., 2020) and that the amount of associated precipitation will increase (Christensen et al., 2013; Zappa et al., 2013; Michaelis et al., 2017; Kodama et al., 2019; Sinclair et al., 2020). However, there is disagreement in how the number of extreme cyclones will change with some studies projecting an increase (Geng and Sugi, 2003; Mizuta et al., 2011; Colle et al., 2013; Pfahl et al., 2015) and some a decrease (Zappa et al., 2013; Michaelis et al., 2017). Despite this, there is a general consensus that the intensity of
the most extreme cyclones will increase in the future (Mizuta et al., 2011; Sinclair et al., 2020; Dolores-Tesillos et al., 2021). Differences in the sign or magnitude of the above changes are thought to be dependent on the choice of model used, the variable examined, the geographic area of interest, and the selection of cyclones studied (see Ulbrich et al., 2009; Catto et al., 2019, for a full review). Consequently, a coherent multi-model analysis of extratropical cyclones on a global scale using consistent methodologies is required to fully understand the changes to cyclone intensity and frequency.


With regards to regional dependence, previous studies have demonstrated that cyclones are projected to increase in intensity in the SH (Chang et al., 2013; Chang, 2017), with a likely decrease in the NH (Zappa et al., 2013; Chang et al., 2013; Colle et al., 2013), with this also being seasonally dependent (Lehmann et al., 2014). Any changes are also dependent on the variable examined (Ulbrich et al., 2009; Catto et al., 2019), with studies that examine mean sea level pressure (MSLP) demonstrating
an increase in cyclone intensity (Bengtsson et al., 2006; Mizuta et al., 2011), and those using vorticity showing little change or even a reduction (e.g. Champion et al., 2011; Zappa et al., 2013; Sinclair et al., 2020). Cyclone wind speed is commonly used to define intensity as it is more directly related to socio-economic impacts, however projections of changes to cyclone wind speeds by the end of the century also tend to be uncertain and vary regionally (Christensen et al., 2013). In a study of NH winter cyclones Bengtsson et al. (2009) and Zappa et al. (2013) found a reduction in the number of cyclones with strong wind speeds,
however, both found increased wind speed intensity over Europe. In composite analyses of North Atlantic cyclones Michaelis et al. (2017) and Dolores-Tesillos et al. (2021) found an increase in wind speeds in extreme cyclones in the region of the warm front. Recently, Sinclair et al. (2020) used an idealized aquaplanet model and also found evidence for an increase in the strength of low-level cyclone wind speeds in the vicinity of the warm front by up to 3.5 m s$^{-1}$. Therefore, an assessment considering a variety of intensity measures is therefore required to clarify these differences.




Commonly only the most extreme cyclones in the top 10% of the intensity distribution are examined as these often pro-
vide the clearest climate change signal (Catto et al., 2010; Sinclair et al., 2020; Chang, 2018, 2017; Zappa et al., 2013) and
are associated with the highest impacts (Ulbrich et al., 2001, e.g.). However, it has been demonstrated in numerous studies that
cyclones in the middle of the distribution can respond differently to the most extreme cyclones (Champion et al., 2011; Pfahl
et al., 2015; Michaelis et al., 2017; Sinclair et al., 2020).

In this study we will utilise a cyclone compositing method (as in Bengtsson et al., 2009; Catto et al., 2010; Dacre et al.,
2012) and a number of models from the new CMIP6 ensemble (Eyring et al., 2016), across a range of future climate scenarios
(O'Neill et al., 2016), to attempt to reduce some of the uncertainty surrounding projected change of cyclone intensity and
cyclone associated wind speeds in the future. Cyclone intensity will be assessed using relative vorticity, MSLP, and cyclone
wind speeds. The questions we address in this study are as follows:

– How will the frequency of extratropical cyclones change in the future and what are the regional variations in changes to
the seasonal storm tracks?

– How is cyclone intensity projected to change and do different intensity metrics give differing projections?

– Do extreme and moderate strength cyclones respond differently to changes in the future climate?

## 2   Data and Methods

### 2.1   CMIP6 Data

In this study the analysis is performed on models that are part of the CMIP6 (Eyring et al., 2016) and ScenarioMIP (O'Neill
et al., 2016) coupled atmosphere-ocean experiments. In total 9 models are used and these are detailed in Table 1. These models
have been chosen as they provide data for the required experiments at the necessary temporal resolutions for tracking, covering
the historical period and the different future scenarios. For investigating sensitivity to future climate forcing four different
experiments are analysed. These are SSP1-26, SSP2-45, SSP3-70, and SSP5-85, and all feature differently evolving climate
forcings throughout the 21st century (see O'Neill et al., 2016, for full details).

The December, January, February (DJF) and June, July, August (JJA) periods are analysed for both the Northern Hemi-
sphere (NH) and Southern Hemisphere (SH), representing each hemisphere's winter and summer season. For the historical
period the years 1979-2014 are used and in each of the future scenarios the years 2040-2100 have been analysed. However, in
all composite analysis only an end of century response is studied and this is the period 2080-2100.





## 2.2 Cyclone Identification and Tracking

For cyclone identification and tracking the method of Hodges (1994, 1995, 1999) is used. In order to identify cyclones the method uses 6-hourly relative vorticity at 850 hPa as an input. As CMIP6 models feature varying atmospheric resolutions, the method first spectrally truncates the vorticity field to T42 and removes the influence of planetary scale waves by masking wavenumbers less than 5. To ensure only mobile, well developed, and long lived cyclones are analysed each track must persist for at least 48 hours and travel at least 1000 km from its point of origin. Furthermore, each cyclone must have a maximum

vorticity of at least $1\times10^{-5}s^{-1}$ (scaled by -1 in the SH). To assign MSLP values to tracks we follow Bengtsson et al. (2009) and use a B-spline interpolation and minimisation technique within a 5° radius from the cyclone center to identify the minimum MSLP value.

## 2.3 Cyclone Compositing

In order to investigate the structural features of cyclones and how they respond to future warming, cyclone composites are

created. The method used here is the same as Bengtsson et al. (2009), Catto et al. (2010), Dacre et al. (2012), Sinclair et al. (2020), Priestley and Catto (2021) and numerous other studies. For the compositing process several intensity thresholds are applied to investigate the changes to different subsets of cyclones. The first subset is on extreme cyclones (EXT), and these must exceed the 90th percentile of peak cyclone vorticity for that hemisphere, season, and model. Secondly, average strength (AVG) cyclones are investigated, and these are situated between the 45th and 55th percentile of the peak cyclone vorticity

distribution. For the historical period the intensity thresholds are calculated across the entire 1979-2014 period, whereas for the future SSPs the thresholds are re-calculated for each decade (i.e. 2040-2049, 2050-2059, etc) to account for the evolving forcing.

For the majority of analysis presented the extreme cyclones around the entire hemisphere are analysed. However, when comparing EXT and AVG cyclones only cyclones that are present over the North Atlantic are studied. These cyclones must have

at least one timestep within a region from 30°N-70°N, 280°E-360°E, which ensures that cyclones are as similar as possible for this comparison.

Composites of the subset cyclone tracks are created at the time of maximum intensity (defined as the T42 vorticity) and cover an area of 20° from the track point at the time of maximum intensity on a 0.5°×0.5° grid. In order to reduce any

discrepancies associated with the different propagation direction of cyclones all composited variables are rotated so that the cyclone is propagating due east. The compositing has been performed on cyclone wind speeds at 850 hPa, 500 hPa, and 250 hPa, to investigate changes in the lower, middle, and upper troposphere respectively. Cyclone wind speed composites are provided in two different perspectives. The first is the Earth relative perspective, which is the winds as output by the models. The second is the system relative perspective, whereby the speed of the cyclone (as calculated from the tracks) is removed and therefore

allows the airflows within the cyclone to be investigated.





## 3  Results

### 3.1  Storm Number

A reduction in the number of cyclones per season is projected in both the winter and summer seasons in the NH and SH (Fig. 1a–d). Across the four seasons the decrease is larger in the simulations that have larger increases in the end of century climate
forcing. In the NH the median decrease in identified cyclones for SSP5-8.5 in DJF (Fig. 1a) and JJA (Fig. 1b) is by 5.4% and 6.8% respectively. In the SH the relative decrease in the SSP5-8.5 experiment is slightly less than in the NH with reductions in DJF (Fig. 1c) and JJA (Fig. 1d) of 3.3% and 4.7% respectively. This result is consistent with the wide array of scientific literature that has assessed changes in cyclone activity in previous generation GCMs (e.g. König et al., 1993; Bengtsson et al., 2009; Chang et al., 2012; Michaelis et al., 2017; Zappa et al., 2013) and can be understood through the reduced lower tropospheric
baroclinicity that is a result of continued Polar Amplification with global warming (Geng and Sugi, 2003; Bengtsson et al., 2006; Catto et al., 2019).

To investigate changes in intense cyclones we define intense cyclones as those that exceed the 90th percentile of the distribution of peak cyclone T42 vorticity. A fixed threshold of the historical 90th percentile is used to assess future changes in the
SSP5-85 experiment. In NH DJF there is an average increase of 4.0% in the number of cyclones exceeding the historical 90th percentile (Fig. 1e). Increases are also seen in the SH with 7.0% and 15.6% more intense cyclones in DJF and JJA respectively (Fig. 1g,h). In NH JJA a decrease is identified, with 21.4% fewer cyclones exceeding the historical 90th percentile (Fig. 1f). It therefore appears that our results for winter cyclones are consistent with previous studies in that a reduction in the total number of cyclones is seen, yet an increase in the frequency of intense cyclones (Geng and Sugi, 2003; Mizuta et al., 2011; Colle et al.,
2013). However, in the summer seasons different behaviour is seen in the NH and SH, with a large reduction in the number of intense cyclones in NH JJA.

### 3.2  Storm Track Density

Despite the decrease in total cyclone numbers projected in Fig. 1a–d, the change in tracks and storm track density can vary significantly on the regional scale. In the NH the maximum cyclone track density is in the two main oceanic storm tracks in DJF
and JJA (Fig. 2a,f), with a further peak in storm track density in the Mediterranean in DJF. The ability of the CMIP6 models in reproducing these features, relative to the most current reanalyses, was discussed in Priestley et al. (2020) and will not be discussed herein.

In DJF the changes in storm track density by the end of the century (2080-2100) are shown for the SSP1-26, SSP2-45,
SSP3-70 and SSP5-85 scenarios in Fig 2b–e. Changes in track density are smallest for SSP1-26 (Fig. 2b) and largest for SSP5-85 (Fig. 2e), which reflects the sensitivity to increasing levels of anthropogenic climate change noted in Fig. 1. All scenarios present a similar pattern of storm track changes.





Looking at the two oceanic storm tracks individually, an apparent poleward shift of the storm track is seen in the North

Pacific with a decrease (increase) in the number of tracks on the equatorward (poleward) flank. This is most evident in the west and central longitudes of the ocean basin, where most of the cyclogenesis and strongest temperature gradients are present (Priestley et al., 2021a). This pattern gets larger in magnitude (and more robust across the models) with the stronger climate change scenarios (Fig. 2e).

In the North Atlantic, there is a decrease in track density in the subtropical central North Atlantic and an increase in the track density over NW Europe and particularly the British Isles. Furthermore, there is a reduction over SW Europe and the Norwegian Seas, therefore presenting a tripolar pattern of change along the Greenwich Meridian. These signals are again largest in the highest emission climate change scenario of SSP5-85, with models also tending to have the highest level of agreement in this scenario. The extension over NW Europe is likely associated with an extension of tracks into this region associated with

increased baroclinicity and higher rates of cyclogenesis (not shown) from modifications to ocean currents in the North Atlantic from the melting of sea ice and ice sheets (Bengtsson et al., 2009; Gervais et al., 2018, 2019; Oudar et al., 2020). The patterns of change in the NH are broadly consistent with the recent studies of Oudar et al. (2020) and Harvey et al. (2020). As in Harvey et al. (2020), our results also demonstrate an increased climate change signal compared to CMIP5, shown by Zappa et al. (2013) and Lee (2015), which used the same cyclone tracking method.


Other regions of note are the Mediterranean and the Arctic, where robust decreases in the number of tracks are observed which are larger with increased future warming. However, due to the often smaller scale of cyclones in these regions, adaptations to identification methods may be required to comprehensively quantify changes in these regions (Zappa et al., 2014, 2015).

In JJA an overall reduction in track density is seen around a majority of the NH, which, as in DJF, is larger for the higher emission climate change scenarios (Fig. 2g–j). For SSP5-85 (Fig. 2j) there is a robust decrease across North America, a majority of Northern Europe, eastern Asia, and the NW North Pacific. The decrease over eastern Asia is co-located with the over-active northern genesis region identified in Priestley et al. (2021a), and is likely driven by changes in the meridional temperature distribution. One region where an increase in track density is identified is to the south, and downstream, of Greenland,

which has been identified in numerous other studies (Bengtsson et al., 2006; Zappa et al., 2013). This is a region where the poleward expansion of the tropical circulation is projected to play a significant role, driving an increase in jet speeds for this region (Harvey et al., 2020). The interaction of the jet with the complex orography of this region is a likely driver of this increase.

The storm tracks in the SH are much more symmetrical in nature (Fig. 3a,f), particularly in the summer season (DJF, Fig

3a). In the summer season (Fig. 3b–e) a poleward shift of the storm track is evident, but only for SSP2-45, SSP3-70, and SSP5-85 (Fig. 3c–e). This indicates some temperature dependency on the poleward shift of the circulation that is not achieved for the lowest warming rates. For the three scenarios that do show a poleward shift, the shift is largest for the scenario with the largest warming, and is symmetric around the hemisphere. For SSP5-85 the average poleward shift of the storm track





relative to the historical simulations (defined as latitude of maximum zonal mean track density) is 2.9° from 55.4°S to 58.3°S.
Furthermore, this shift is a very robust signal across all the models, with the most widespread agreement in SSP5-85. Another
feature of note is the robust decrease in storm track density from the sub-tropical western South Pacific, to the southern tip of
South America, which may be associated with changes in the South Pacific convergence Zone (SPCZ; Brown et al., 2020).

In the JJA (Fig. 3g–j), the patterns of track density change are very similar to those of DJF. There is a poleward shift of
the storm tracks that is only seen for SSP2-45 and above, with a signal that is larger for the largest climate change scenario.
This shift is also robust across the models. The poleward shift is smaller than in DJF and for the SSP5-8.5 scenario the average
poleward shift from the historical simulations is by 1.3° from 60.8°S to 62.1°S. The clear pattern in the SH is likely due to
the reduced influence of land in this hemisphere, therefore allowing for the strong thermodynamical changes to have a clearer
impact. The poleward shift of the circulation is expected through increased tropical warming and an expansion of the Hadley
Cell (Shaw, 2019), which results in an increase in stability in subtropical regions and a poleward shift of the main baroclinicity
(Yin, 2005; Lim and Simmonds, 2009). Furthermore, the amplification of polar temperature in the NH is not projected to be as
large in the SH (Fan et al., 2020), further maintaining the mid-latitude baroclinicity.

### 3.3 Cyclone Intensity

We have already shown that, as measured through relative vorticity, there is a projected increase in the number of extreme
cyclones in the future (Fig. 1e–h). However, to quantify differences in measures of cyclone intensity, both the T42 relative
vorticity, and associated cyclone MSLP minimum will be examined. For both measures the peak intensity is defined as the
maximum (minimum for MSLP) value throughout the cyclone lifecycle.

### 3.3.1 Relative Vorticity

Figure 4 shows the peak cyclone relative vorticity distributions for DJF and JJA in the NH and SH. For all seasons, and all future
experiments, the model mean distributions lie within the uncertainty estimation of the historical models, indicating a relatively
small change in cyclone vorticity. Estimations of the median vorticity are very similar for all of the future SSPs relative to the
historical models in the winter seasons. However, in the summer seasons a decrease is evident, with this being notably larger in
the NH.

All distributions exhibit a similar shape for the SSPs as in the historical simulations. Despite this, all seasons except NH
JJA show a reduction in the number of cyclones near the peak of the distribution. This confirms that the reduction in cyclones
noted in Fig. 1 is most likely to be seen for moderate strength cyclones and not extreme cyclones (as in Fig. 1). Since a key
question is how the intensity of the highest impact storms will change, the changes in the 90th percentile of the vorticity
distributions are shown in the inset of each panel in Fig. 4. The model mean 90th percentile increases slightly for NH DJF and
both SH seasons, yet decreases for NH JJA. However, the uncertainty estimation on the 90th percentile for each SSP overlaps
the median estimation from the historical models. Therefore, there are different responses in the median and 90th percentile for





the winter seasons and SH summer, with extreme cyclones increasing in intensity with no change (or a slight decrease) in the median intensity.

### 3.3.2 Mean Sea Level Pressure

In Fig. 5 distributions of cyclone minimum MSLP are shown and calculated the same way as for relative vorticity in Fig. 4. The MSLP distributions have quite different shapes, with the NH DJF distribution having a negative skew (Fig. 5a) and NH JJA being normally distributed (Fig. 5b). In NH DJF (Fig. 5a) the distributions of the four SSPs are similar to the historical models, but with a reduction in the number of cyclones with MSLP at the peak of the distribution. There is a shift toward lower pressures in NH DJF, with lower medians and also a lower 90th percentile (inset). The medians show more of a shift than
for vorticity, although the 90th percentiles show less of a shift (Fig. 4a). In NH JJA there is a slight shift toward lower intensities (higher MSLP), although the distributions are again very similar with less of a shift in the median than for relative vorticity.

In the SH the distributions of MSLP in the SSPs is largely different compared to the historical experiment. In DJF (Fig. 5c) there is a large reduction in frequency of cyclones with average or above average MSLP, leading to a reduction in the
median. Furthermore, there is an increase in cyclones with the lowest pressures, resulting in a lower 90th percentile and the average distribution nearly lying outside the uncertainty from the historical simulations. These shifts in the overall distribution, the median, and the 90th percentile are also apparent in JJA (Fig. 5d), with a shift toward lower pressures (higher intensity) in the SSPs that is exacerbated in the highest emission scenarios. This is unlike the distributions of relative vorticity (Figs. 4c,d), which show only an increase in the 90th percentile of the distribution and minimal changes (or a slight weakening) in the median.

For MSLP there are larger changes in the median of the distribution than for relative vorticity, with this being particularly notable in the SH. This is likely a result of the poleward shift of cyclones and the significant influence of the large-scale pressure distribution on cyclone MSLP (Bengtsson et al., 2006). As cyclones shift toward higher latitudes (Fig. 3) they will be moving into an environment characterised by deeper MSLP and stronger pressure gradients. This is also likely the reason why less
change is seen in the 90th percentile of MSLP in NH DJF when compared to relative vorticity, as the change in latitude of maximum track density is less apparent than in the SH (Fig. 2a–e).

### 3.4 Cyclone Wind Speed

Figures 4 and 5 illustrate the difference in intensity measures, with changes in the extremes more notable using filtered vorticity as a metric, and a larger shift in the whole distribution when using MSLP. A dynamical approach to assessing changes in
extratropical cyclones strength is to analyse changes in wind fields, as this is a metric that is less influenced by the large-scale environment. For the composite analysis only the 10% strongest cyclones are examined as these are associated with the largest impacts and to ensure only similar cyclones are examined.





### 3.4.1  Lower Troposphere

Recently, Priestley and Catto (2021) demonstrated that CMIP6 models under-estimate the strength of the system relative circu-
lation at 850 hPa by up to 2 m s$^{-1}$. This under-estimation was identified on the poleward flanks of the cyclone in the region
where the CCB would be expected to be found. The models used in this analysis show similar biases, albeit slightly smaller in
magnitude (Fig. S1), due to the increased number of higher resolution models used (Table 1).

The future response of the 850 hPa wind speeds in extreme cyclones is examined for each winter/summer season in both
hemispheres individually and for both a system and Earth relative perspective. Fig. 6 presents the climate change responses for
NH DJF. In a system relative perspective a strengthening of the wind speeds is seen for all SSPs (Fig. 6b–e). This strengthening
is broadly contained between 5° and 15° of the cyclone centre and is progressively larger for the strongest climate change
scenarios with a peak increase of up to 0.7 m s$^{-1}$ in SSPs 3-70 and 5-85. The strengthening is robust across most of the models,
especially for SSPs 3-70 and 5-85 (Fig. 6d,e) and is concentrated on the forward and rearward flanks of the cyclone. The robust
strengthening to the west of the cyclone centre in SSP5-85 is co-located with the maximum wind speeds associated with the
CCB and formation of the low-level jet (Priestley and Catto, 2021). It is evident from SSPs 2-45 and 3-70 (Fig. 6c,d) that there
is also a strengthening on the SE edge of the cyclone in the warm sector, indicating possible changes associated with the inflow
of the WCB.

In an Earth relative perspective (Fig. 6f–j) the progressively stronger increase noted from Fig. 6b–e is also evident, with
SSP5-85 having the greatest strengthening of the wind speeds. From this perspective there is a notable increase in strength on
the southern and southeastern flanks of the cyclones, which would be situated within the warm sector of the cyclones. These
anomalies are 10°-15° away from the cyclone centre and peak at up to 0.9 m s$^{-1}$ for SSP5-85 (Fig. 6j). As the differences
between the system and Earth relative perspective comes from the addition of the speed of the cyclone, this suggests that
cyclones will also be moving faster in a future climate. This increased speed, coupled with a strengthening of system wind
speeds, may lead to increased wind impacts.

In NH JJA (Fig. 7) in the very core of the cyclone a strengthening of system relative winds of up to 0.9 m s$^{-1}$ is seen
for SSP5-85 (Fig. 7e), with a weakening identifiable outside this between 5° and 10° of the cyclone centre of up to 0.7 m s$^{-1}$
Priestley and Catto (2021) and may be associated with the CCB. The weakening is concentrated on the forward flank of the
cyclone, and for the more aggressive climate change scenarios (Fig. 7d,e) in the warm sector. All anomalies are most notable
for SSP2-45 and above and they also become more robust across the models with larger climate change. The reduction in wind
speeds surrounding the core of the cyclone are likely a result of the reduced cyclone pressure gradient (not shown).

In an Earth relative perspective (Fig. 7g–j), a similar pattern of the biases is seen as in the system relative perspective,
however a further weakening is evident on the southeastern flank of the cyclone that is between 5° and 10° of the cyclone centre





and is by up to 0.9 m s$^{-1}$ in SSP5-85 (Fig. 7j). This further weakening suggests that (as with the strengthening in the same location in NH DJF) that there are changes in cyclone speed, and therefore a slowing down of cyclones in NH JJA in the future, with this being progressively larger with more aggressive climate change. It is interesting that the response of cyclones in the NH is oppo-
site in DJF and JJA, which is consistent with the changes in eddy kinetic energy and baroclinicity noted by Lehmann et al. (2014).

The SH winter (JJA, Fig. 8) is consistent with NH winter (Fig. 6) in that a strengthening of the cyclone wind speeds is seen. In a system relative perspective (Fig, 8a–e) a minimal increase is seen for SSP1-26 (Fig. 8b). However, for the remaining SSPs an increase is seen with the largest increases on the poleward flank of the cyclone centre (Fig. 8c–e). This strengthening is
located in a similar location to the underestimation noted in the historical simulations relative to ERA5 (Fig. S1 and Priestley and Catto, 2021), which are likely associated with the CCB. In SSPs 3-70 and 5-85 there is also a strengthening on the equatorward side of the cyclone, which may be a result of the stronger pressure gradients (not shown). Furthermore, the largest strengthening is generally confined to within 4° and 8° of the cyclone centre, which is further away from the centre than the largest biases in the historical simulations, and also further away from the centre than the strongest wind speeds in Fig. 8a), indicating a pos-
sible expansion of the wind field. The strength of the increase is up to 1.8 m s$^{-1}$ and is larger than in NH DJF in SSP5-85 (Fig. 8e).

In an Earth relative perspective, the cyclones in SH JJA (Fig. 8f–j) exhibit a large and robust strengthening on the equatorward flank of the cyclone. Unlike in the system relative perspective, the Earth relative anomalies get progressively stronger with each SSP, resulting in widespread anomalies of over 1.8 m s$^{-1}$ in SSP5-85 (Fig. 8j). As in both seasons in the NH,
differences between the Earth relative and system relative perspectives allow for differences in the cyclone propagation speed to be ascertained. These anomalies provide evidence that (as in NH DJF) the cyclones are moving faster in SH JJA.

As with the vorticity and MSLP distributions (Fig. 4, 5), the lower tropospheric wind speed response in the SH summer (Fig. 9) is consistent with the response from the winter for extreme cyclones (Fig. 8). In both a system relative and an Earth
relative perspective the circulation strengthens for the larger climate change scenarios, with minimal changes identifiable in the SSP1-26 simulations (Fig. 9b,g). In the system relative perspective the wind speeds increase on the poleward flank of the cyclone, with the largest change being between 5° and 10° from the cyclone centre, which is outside the maximum from the historical simulations (Fig. 9a) and as in JJA is suggestive of an expansion of the circulation. The largest increase in system relative winds is by up to 1.4 m s$^{-1}$ for SSP5-85 conditions (Fig. 9e). As with all other seasons in both hemispheres
the change in wind speeds in the Earth relative perspective is concentrated on the equatorward flank of the cyclone for SH JJA (Fig. 9f–j). The wind speed changes are larger for the stronger climate change scenarios and peak at up to 1.4 m s$^{-1}$ under SSP5-85 conditions (Fig. 9j). As with all other seasons, these increases demonstrate a change in the cyclone propagation speed, with strengthenings in the Earth relative perspective, that are not present in the system relative perspective in SH JJA.

It is notable from the lower tropospheric composites (Figs. 6–9) that the responses of extreme cyclones to climate change in the NH and SH summer seasons are again different, as they were in the vorticity and MSLP distributions (Fig. 4, 5). This





response is expected from the enhanced pressure gradients in the SH (not shown) and the persistent poleward shift of the storm tracks in the SH in both seasons (Fig. 3). This is a contrast to the NH summer, whereby the storm tracks do not shift poleward, and instead a reduction in cyclonic activity is seen around the entire hemisphere (Fig. 2f–j). This is likely due to the larger
polar amplification noted in the NH and reduction in lower tropospheric baroclinicity across the hemisphere, which has a strong impact on the NH storm tracks and general circulation (Coumou et al., 2015; Harvey et al., 2015).

### 3.4.2 Cyclone Extreme Wind Footprint

For all seasons the largest 850 hPa Earth relative winds are commonly found within 5° of the cyclone centre (Fig. 6–9). The largest future changes, however, are often found further from the cyclone centre, and can be 10°-15° from the cyclone centre.
For all seasons it is likely that there are changes in how broad the circulation is. To quantify the change in the broadness a transect of the circulation is taken on the equatorward side of the cyclone, from the cyclone centre to the edge of the composite area, thereby intersecting the region of maximum wind speeds (dashed white line Fig. 6f). The fraction of this transect above a fixed wind speed threshold is calculated and plotted against the maximum wind speed of the cyclone along the same transect. Wind speed thresholds have been chosen to best represent the footprint of the historical composites.


For NH DJF (Fig. 10a) there are only small changes in maximum wind speed, however there are increases in the broadness of the area of above threshold wind speeds, with the peak of the distribution being ~0.5° broader in the SSP5-85 experiment than in the historical period. Conversely, in NH JJA (Fig. 10d) there is a considerable contraction of the area of above threshold wind speeds by nearly 2° that is also associated with a lower maximum wind speed. In the SH we find an expansion in the area
of above threshold wind speeds in both DJF and JJA (Fig. S2). However, unlike in the NH, this is also associated with a slight increase in the maximum wind speeds at 850 hPa.

As the broadness of the circulation is increasing (in NH DJF) we can quantify the area of the cyclone above a fixed threshold for the entire composite area and how this evolves for the different SSPs over time. Figure 11 shows timeseries of the winter and summer seasons in the NH and SH for the area of the cyclone composite above a fixed threshold of 17 m s$^{-1}$. The differing
responses in the DJF and JJA in the NH are notable in Fig. 11a,b with the SSPs showing an increase in cyclone area above 17 m s$^{-1}$, and a decrease in JJA, by the end of the century. The differences between the SSPs are not overly large in the NH, with SSP5-85 showing a median increase of 7% of above threshold winds by 2100 in DJF, which is similar to SSP3-70 (Fig. 11a). However, increases may be as large as 15% in SSP5-85 based on the large spread of model projections, with almost
half of the SSP3-70 and SSP5-85 estimations exceeding the historical 75th percentile in the final decade of the 21st century. The SSP1-26 and SSP2-45 scenarios generally show lower increases in the area of above threshold wind speeds with median increases of 2-3% by the end of the century. All the SSPs show a very similar average evolution until 2080 of a 2.5-5% size increase, with the two most aggressive SSPs only separating from the two weakest SSPs after ~2080. Interestingly, the SSP2-45 scenario indicates a decline after 2075, which is in line with modelled reductions in $CO_2$ emissions toward the end
of the 21st century in that SSP (O'Neill et al., 2016). Despite the differences in scenarios, all of the SSPs in Fig. 11a have a




median extreme wind speed area lower than the 75th percentile from the historical simulations by 2100. In JJA all SSPs are very similar and generally show up to a 10% reduction the area of above threshold winds throughout the century. Generally, SSP5-85 and SSP3-70 show the largest reduction, which may be as large as 25-30%. Despite this, the inter-annual variability is especially large for all SSPs and each has considerable overlap with a large amount of the historical distribution of cyclone area.


In the SH a larger signal, with greater differences between the SSPs, is identifiable for the area of above threshold cyclone winds (Fig. 11c,d). For both DJF and JJA the higher emission SSPs demonstrate a marked increase in cyclone area of above threshold winds relative to the historical simulations. In DJF (Fig. 11c) the SSP5-85 simulations has an extreme wind speed area nearly 15% larger by the end of the century, which is also considerably above the 75th percentile from the historical simulations. The SSP3-70 simulation demonstrates around a 10% increase by the end of the century, which is comparable to the 75th percentile of the historical simulations. Both the SSP5-85 and SSP3-70 simulations are similar until around 2075, when the rate of increase in cyclone area levels off in SSP3-70. Accounting for the spread in the model simulations by the end of the century indicates increases may be as large as 20-25% relative to the historical average, with a majority of the SSP3-70 and SSP5-85 distributions being above the historical 75th percentile. The SSP2-45 experiment is very similar to SSP3-70 until 2070, however the average area increase is lower at only 5% above the historical average by the end of the century. The SSP1-26 experiment demonstrates the smallest by the end of the century, which is within the range of 75th percentile and less than a 5% increase in area compared to the historical simulations. The SSP1-26 simulation actually increases until approximately 2070, when it closely matches the SSP2-45 and SSP3-70 scenarios, with a decline then following, before levelling off by approximately 2085. By the end of the century the spread in the SSP1-26 simulations is very similar to that of the historical cyclone area.


In SH JJA (Fig. 11d) a similar evolution to DJF (Fig. 11c) is notable, albeit with a larger signal in the SSP5-85 and SSP3-70 scenarios. Both SSPs demonstrate large increases in cyclone area above the threshold wind speed compared to SSP2-45 and SSP1-26 and show an increase in an average area increase of at least 20% by the end of the century. From 2090-2100 a majority of the distribution in both SSPs exceeds the 75th percentile of the historical simulations, and the 75th percentile of the SSP5-85 scenario estimations are 35-40% larger than the historical average. For both these SSPs the cyclone area demonstrates a steady increase from 2040, by which all experiments already show an increase in size of 5-10% above the historical average. Neither SSP2-45 or SSP1-26 show a marked increase from 2040-2100. SSP2-45 maintains an average of approximately 10% increase in cyclone area by 2100, whereas SSP1-26 shows a very slight decrease to an average of a 5% increase in size by 2100. The clearer signal in the SH and greater separation in the SSPs is (as mentioned previously) is likely due to the reduced land influence in the SH allowing for a more robust response to the evolving climate forcings. In fact, the evolution of the cyclone size above the 17 m s$^{-1}$ wind speed threshold bears considerable similarity to the evolution in $CO_2$ concentration noted in the ScenarioMIP experimental design (O'Neill et al., 2016, Fig. 3b).





### 3.4.3 Middle Troposphere

The middle troposphere is defined as 500 hPa in this analysis. In the interest of brevity only the NH DJF biases will be discussed
in detail, with changes from other seasons available in the supplementary material (Fig. S3–S5) and discussed briefly at the end
of this section.

In NH DJF (Fig. 11) the spatial patterns of change across the models presents a similar pattern across all SSPs in both a
system relative (Fig. 12b–e) and an Earth relative (Fig. 12g–j) perspective. In the system relative perspective there is first of
all a strengthening of the wind speeds on the poleward flank of the cyclone centre that is ~5° away from the centre. This is a
region where winds are travelling from east to west (Fig. 12a). This element of the circulation in the mid-troposphere is likely
associated with the upper reaches of the CCB (Priestley and Catto, 2021) and this may indicate a strengthening of this airstream,
which is robust across a majority of the models.

The other notable difference in the system relative winds is situated within the warm sector of the cyclones. For all SSPs
there is evidence of a strengthening of the winds, which is progressively stronger with the more aggressive climate change
scenarios and by up to 1 m s $^{-1}$ in SSP5-85 (Fig. 12e). This increase is located between 5° and 10° of the cyclone centre
on the equatorward and eastern flanks. These regions are associated with the regions of strongest ascent (Catto et al., 2010;
Priestley and Catto, 2021) and therefore indicates a possible strengthening of the WCB. A large amount of the strengthening
in the warm sector is due to an increase in the meridional component of the wind by over 1.8 m s$^{-1}$ to the east and southeast
of the cyclone centre (not shown). It is likely that associated with this stronger meridional motion is an increase in the ascent
rate of the cyclones, however due to the limited data output of some CMIP6 models, a full assessment of this is not possible.
This stronger warm sector motion is likely a result of increased atmospheric moisture content under future climate conditions,
which will result in larger condensational heating in ascending branches of cyclones and potentially increased rates of ascent.

One final feature to note across all the SSPs in the system relative framework is the weakening of the circulation on the
rearward (western) flank of the cyclone between 5° and 15° of the cyclone centre. This region is behind the cold front and
generally is associated with large scale descent of drier air (Catto et al., 2010; Priestley and Catto, 2021). This weakening
suggests that the descent rate may be weaker under future climates, which will again be exacerbated with larger levels of climate
change.

In the Earth relative perspective (Fig. 12g–j) a very similar picture is presented as in the system relative perspective (Fig.
12b–e). The main difference is the increased strength of the changes in the warm sector in the Earth relative perspective, that is
also slightly broader than in the system relative perspective. In Fig. 10b it can be seen that, as at 850 hPa, the area of extreme
wind speeds becomes considerably broader. However, unlike at 850 hPa the maximum wind speed also increases at this level.
This may be due to a larger role of moist processes in the mid-troposphere, however projected increases in the strength of the



upper-level jet (Harvey et al., 2020) may also have an influence at this level.

In the NH JJA (Fig. S3) and both SH seasons (Figs. S4–S5) the pattern of change is broadly consistent with NH DJF
(Fig. 12), but with magnitudes and sign that are consistent with the biases noted at 850 hPa (Figs. 7–9). For NH JJA (Fig. S5)
there is weakening of the wind speeds in the warm sector of the cyclones which is progressively larger for the more aggressive
SSPs. Furthermore, the broadness of the extreme wind speeds also decreases (Fig. 10e). In the SH the responses in both seasons
are very similar (Figs. S4–S5), unlike in the NH. Both SH seasons feature a pronounced strengthening of the wind speeds in
the warm sector with the largest anomalies being between 5° and 15° of the cyclone centre on the equatorward and forward
flanks of the cyclone. For all seasons the changes in the warm sector have larger magnitude in the Earth relative perspective,
again suggesting changes in cyclone speed in the future.

Both SH seasons also show a strengthening of the wind speeds on the poleward flank of the cyclone in the system rela-
tive framework, at a distance of approximately 5°-7°, which is likely associated with the strengthening of the CCB noted
earlier.

### 3.4.4   Upper Troposphere

As in the previous section, only the biases for NH DJF (Fig. 13) will be discussed in detail due to the similar responses of the
other seasons and magnitude that is consistent with anomalies at the other tropospheric levels. Biases for the other seasons are
present in Figs. S6–S8.

The changes in the wind speeds at 250 hPa in NH DJF can be broadly split into 3 distinct features in the system relative
perspective (Fig. 13a–e). Firstly, 10°-20° to the southwest of the cyclone there is an increase in the wind speeds, which is larger
for the strongest SSPs. This is likely associated with a strengthening/broadening of the upper-level jet (Fig. 10c). Secondly,
there is an increase in the wind speeds by up to 2.1 m s$^{-1}$ to the south, east and northeast of the cyclone centre that is ~5°
from the cyclone centre and rotating cyclonically around it. This is the region of largest ascent at 250 hPa and the region where
models struggle to capture the strength of the wind speeds in historical simulations (Priestley and Catto, 2021). This increase is
situated above and slightly downstream of a similar increase at 500 hPa (Fig. 12b–e) and therefore likely part of the ascending
feature that moves in a slantwise manner in the warm sector. This strengthening is mostly a result of stronger meridional motion
(as at 500 hPa) and together suggests that the WCB in cyclones is likely to be stronger, and progressively so with increased
warming rates by the end of the century.

The third change at 250 hPa is an increase in wind speeds on the forward flank of the cyclone between 10° and 20° of
the cyclone centre and extending in a southerly and southwesterly direction. The strengthening is mostly driven by a more
negative meridional component of the wind, which is associated with the anticyclonic turning of air and likely part of the WCB
outflow. We hypothesise that this strengthening is result of the stronger motion near the core of the cyclone, which is ascending





at an increased rate and having a larger divergence when it reaches the tropopause.

All these anomalies are largest for SSP5-85 (Fig. 13e) with strong model agreement across the SSPs, indicating that the warmer climate, which is characterised by a higher moisture content in the future, results in a stronger wind speeds in the warm

sector and anticyclonic turning near/at the tropopause. In addition, there is evidence of weaker wind speeds to the northwest of the cyclone and behind the cold front at 250 hPa, which is consistent with the weakening also noted at 500 hPa (Fig. 12) and suggests that the region of descent within the cyclones, which often originates near this level, is likely to be weaker in the future.

In an Earth relative perspective (Fig. 13f–j) a majority of the features noted in the system relative perspective (Fig. 13a–e) are also notable and consistent in their magnitude and positioning. As at other levels an increase in the strength of change, particularly on the equatorward flank of the cyclones, is notable, which is a result of the increasing speed of cyclones in the future. As in the lower and middle troposphere a considerable broadening of the circulation is also identifiable at 250 hPa (Fig. 10c) alongside an increase in the maximum speed to the south of the cyclones.


Composites for the changes in the 250 hPa wind speeds for the NH JJA (Fig. S6) and also for DJF and JJA in the SH (Figs. S7–S8) are now discussed briefly. For these other seasons a majority of the same features identified for NH DJF (Fig. 13) are seen, with magnitudes of changes that are consistent with the same season in the middle and lower troposphere. In NH JJA a weakening of the wind speeds is again identifiable throughout the warm sector of the cyclone. Due to the considerably weaker

nature of the wind in NH JJA the separate changes associated with the anticyclonic motion is harder to identify, however, the meridional component of wind shows a marked increase to less negative values (weakening, not shown) to the southeast of the cyclone indicating a consistent (yet opposite) behaviour to the change in NH DJF.

In the SH both seasons (Figs. S7–S8) are very consistent with NH DJF in that strengthening wind speeds associated with

both the cyclonic and anticyclonic motions within the composite area are identifiable, with both only becoming evident for SSP2-45 and above. For both seasons in the SH, and NH JJA the changes are largest in magnitude for SSP5-85 and progressively lower for each SSP below. All of the major changes are robust across the model ensemble and represent a clear picture as to how cyclones will change as a result of future climatic change.

For all levels in the troposphere the cyclones in the SH only start to show a consistent signal and pattern of change for SSP2-45 and above, whereas the pattern of the anomalies in the NH is usually evident (although often small in magnitude) even for SSP1-26.





### 3.5 Sensitivity of Change to Cyclone Intensity

We have already shown that extreme cyclones increase in intensity for relative vorticity and MSLP (decrease in NH JJA; Fig.
4, 5). However, it is only for MSLP that an increase in intensity of moderate strength cyclones is seen when the extreme
cyclones are also intensifying (Fig. 5). To assess how cyclones from different parts of the intensity distribution are changing dy-
namically, 850 hPa wind speeds from cyclones that are present over the North Atlantic (30°N-70°N, 280°E-360°E) are examined.

The wind speeds at 850 hPa for the AVG cyclones (Fig. 14a,d) have a much smaller footprint than the EXT cyclones (Figs. 6–9).
Furthermore, wind speeds are lower, with the maximum being less than 20 m s$^{-1}$. For the EXT cyclones the largest changes in
system relative wind speed in the SSP5-85 experiment are on the southeastern flank of the cyclone and approximately 5°-10°
from the cyclone centre (Fig. 13c,f). The AVG cyclones in the North Atlantic exhibit a similar sign of change, although the
magnitude is smaller (Fig. 14b,e) and, in the case of DJF, focussed in a different part of the cyclone.

In DJF, the AVG cyclones (Fig. 14b) feature a strengthening of up to 1 m s$^{-1}$ to the south and southwest of the cyclone
centre at a distance of approximately 10°. The increases to the southeast of the cyclone centre, which are concentrated in the
warm sector of the cyclone, are absent or smaller in magnitude that the EXT cyclones (Fig. 14c). Furthermore, changes in
the system relative wind of AVG cyclones is considerably smaller and slightly negative to the southeast of the cyclone centre,
whereas this is positive and concentrated in the warm sector for EXT cyclones (Fig. S9). Therefore, the changes observed in
AVG cyclones are largely driven by changes in the cyclone speed, whereas changes in EXT cyclones involve changes to strength
of circulations within the cyclone, as well as an increase in translational speed.

In JJA, a weakening of the 850 hPa circulation is present in both AVG and EXT cyclones (Fig. 14e,f). For both cyclone
groups the largest changes are on the southern flank of the cyclone, however they are larger for the EXT cyclones (as in DJF).
It therefore appears that the circulations of EXT cyclones tend to respond more to changes in future climate forcing and have a
larger magnitude of change than AVG cyclones. A similar pattern is also seen in the SH and for cyclones present over the North
Pacific (not shown).

## 4    Conclusions and Discussion

### 4.1    Summary

This paper presents the first assessment of changes to objectively identified cyclones, their intensities, and associated wind
speeds in the newest generation CMIP6 models. We have also considered a number of metrics to assess changes in intensity
across a range of different shared socio-economic pathways and the main findings of our work are as follows:





- Cyclone numbers are projected to decrease globally in both winter and summer seasons. This decrease will be largest with the greatest increase in climate forcing (Fig. 1). Extreme cyclones are likely to be more common in winter seasons, with more cyclones exceeding historical intensity thresholds.

- In the Northern Hemisphere winter there is a projected poleward shift of the North Pacific storm track, an extension of the North Atlantic storm track into Europe, and a decrease in activity over the Mediterranean. In summer a reduction in storm track activity is projected hemispherically. For both seasons the projected changes increase with the more aggressive climate change scenarios (Fig. 2).

- In the Southern Hemisphere a robust poleward shift of the storm tracks is projected for both seasons in a symmetric manner around the hemisphere (Fig. 3).

- The most extreme cyclones will have higher intensities (higher vorticity/lower MSLP) for both seasons in the SH and for NH winter, whereas the strongest summer cyclones in the NH will weaken slightly (Fig. 4, 5).

- The strongest cyclones will have stronger wind speeds throughout the troposphere for NH winter and both seasons in the SH. The largest increases are noted in the warm sector of the cyclones (Figs. 6, 8, 9, 12, 13).

- The area of extreme wind speeds will increase in future winters (Fig. 10) and the area of intense wind speeds in extreme cyclones will increase by up to 40% by the end of the century under SSP5-85 conditions (Fig. 11).

- Extreme cyclones are projected to intensify at a larger rate than average strength cyclones in the winter (Fig. 14).

Our results have shown that winter extratropical cyclones are projected to increase in intensity and severity with increased global warming in both the NH and SH. The largest changes are seen in experiments that feature the largest rate of global warming (SSP5-85). However, the most extreme changes, and resultant impacts, can be avoided through limiting the amount of warming experienced. Under SSP2-45 conditions the area of extreme wind speeds in winter cyclones increases by less than half the amount in SSP5-85 or SSP3-70, with only minimal differences relative to the historical distribution.

The pattern of change in objectively identified cyclones in the CMIP6 models is consistent with studies from previous generation CMIP5 models (Mizuta, 2012; Zappa et al., 2013; Lee, 2015). Changes in the structure of the storm tracks in the NH are also consistent with changes in the eddy driven jet (Oudar et al., 2020) and the Eulerian measure of storm track (Harvey et al., 2020). In the SH the identified robust poleward shift of the storm track is consistent with previous studies (Yin, 2005; Chang et al., 2012) and analysis of the SH jet and Southern Annular Mode (Goyal et al., 2021; Grose et al., 2020). The poleward shift of tracks in the SH and North Pacific is likely associated with changes in latitude of maximum baroclinicity (Yin, 2005) and also increasing levels of atmospheric water vapour (Tamarin and Kaspi, 2017; Tamarin-Brodsky and Kaspi, 2017). The increased magnitudes of change in both hemispheres relative to CMIP5 (Lee, 2015) are likely a result of the larger temperature changes in CMIP6 models relative to CMIP5 (Tebaldi et al., 2021) as a result of stronger equilibrium climate sensitivity (Zelinka et al., 2020).






We have identified increases in wind speeds in the warm sector of the cyclones in the winter seasons, where the warm conveyor belt is dominant. This is somewhat consistent with Sinclair et al. (2020) and Dolores-Tesillos et al. (2021) where changes in cyclone lower tropospheric wind speed was identified in the vicinity of the warm front, although the changes found by Sinclair et al. (2020) are on the forward flank of the cyclone and the response was nearly twice as large as identified here. The

processes driving the increases (or decreases) in wind speed throughout the troposphere would require further investigation, however, the intensification, and ascent rate, of extratropical cyclones are intrinsically connected to the WCBs (Binder et al., 2016), which are likely to undergo modification under future climate change with additional atmospheric water vapour (Tamarin-Brodsky and Kaspi, 2017) and diabatic processes (Dolores-Tesillos et al., 2021) and should be an area of future scientific interest.

A large amount of the change in wind speed in the Earth relative framework appears to be due to changes in the propagation speed of cyclones. In seasons where an intensification is seen the cyclones appear to travel faster, and vice versa. Previous studies have documented both an increase (Geng and Sugi, 2001; Jiang and Perrie, 2007), and a decrease (Löptien et al., 2008) in propagation speed in the future, which may be connected to changes in the speed of the mid-latitude jets (Barnes and Polvani, 2013), however, further investigation would be required to determine the exact processes driving the changes in cyclone speed.


We have shown how the area of a cyclone above a fixed wind speed threshold is projected to increase in the future, with an increase in area of up to 25% noted during SH JJA by the end of the century. This has significant implications for loss modellers as extratropical cyclones are already the main driver of losses in the mid-latitudes (Schwierz et al., 2010). This increase in strong wind area could contribute to the projected increase in European windstorm losses in the future (Pinto et al.,

2007; Leckebusch et al., 2007).

## 4.2 Discussion of metrics

It has been suggested in numerous studies (e.g. Ulbrich et al., 2009; Catto et al., 2019) that the projected changes in cyclone intensity depend on the variable examined, the definition of intensity, and the subset of cyclones examined. Here we have used a number of measures to define intensity; central maximum MSLP and 850hPa vorticity, numbers of cyclones above a fixed MSLP

or 850hPa threshold, extent of low-level strong winds around the cyclones, and strength and extent of high winds at different levels of the atmosphere. We have shown that each of these measures gives the same conclusions. Extreme cyclones have higher intensities (lower in NH JJA) regardless of the intensity metric examined (consistent with Lambert and Fyfe, 2006; Mizuta, 2012; Colle et al., 2013; Zappa et al., 2013; Chang, 2017; Michaelis et al., 2017; Sinclair et al., 2020). Therefore, for examining cyclone extremes, most metrics will provide similar conclusions, giving robust projections of more impactful storms in the future.


We have also investigated two different subsets of the cyclones; the top 10% of cyclones in terms of their vorticity (EXT), and the 10% of cyclones from the middle of the distribution (AVG). For cyclone wind speeds, the AVG cyclones do not see as large of a change in their wind speeds as the EXT cyclones, which is consistent with Michaelis et al. (2017) and Sinclair et al.





(2020). There are differences in the magnitudes of change between those studies and our results which may be a result of chosen

geographical region, with Michaelis et al. (2017) focussing just on the North Atlantic, or the model setup, with Sinclair et al. (2020) based on aquaplanet experiments. We find changes in MSLP for both AVG and EXT cyclones (particularly in the SH; Fig. 5). This is due to the large influence of the latitude change on AVG cyclones and the climatologically lower pressures at higher latitudes (Bengtsson et al., 2006). MSLP is more influenced by the large-scale background state (Hoskins and Hodges, 2002), and is a feature that models struggle to represent (Priestley and Catto, 2021). It is therefore recommended that overall

changes in cyclone intensity are not assessed using MSLP, and instead variables that are less influenced by the large-scale atmospheric state, such as winds or vorticity, are used.

### 4.3 Limitations and Future Directions

Further to the results presented in this study there are several caveats to the work we have presented and consequently areas that could be investigated further. Due to the limited number of models that have made available experiments across the four

SSPs we have investigated, we were limited to just 9 models. A larger ensemble size would be beneficial in order to reduce the model variability and is something that may be possible in the future with the continuous release of new CMIP6 model data. Furthermore, we have focussed on all identified cyclones that exceed a hemispheric threshold and not investigated any regional changes. Narrowing down our research area regionally would be especially beneficial in the NH where the North Atlantic and North Pacific storm tracks show different responses to increased levels of climate change, and may explain the differences seen

between our results and those of other studies (e.g. Michaelis et al., 2017; Sinclair et al., 2020). There is also evidence that higher resolution models show larger increases in cyclone storminess in the future (Grist et al., 2021), and a composite analysis of HighResMIP cyclones would allow for a deeper analysis of how the cyclones in high resolution models evolve differently to in low resolution models.

Moving forward a wider array of physical properties would be very useful to investigate to explore the more detailed aspects of extratropical cyclones. Precipitation and vertical velocity would be interesting variables to explore which have been shown to increase for cyclones in future climates (Kodama et al., 2019; Sinclair et al., 2020) and would allow for a more thorough diagnosis of the changes we have identified, however investigation such as this in CMIP6 models is significantly limited by the insufficient temporal resolution of the available data and limited output of required variables. Finally, despite

significant improvements in CMIP6 model capabilities (Priestley et al., 2020; Bracegirdle et al., 2020; Tian and Dong, 2020; Davini and D'Andrea, 2020), model biases still remain. In CMIP6 models a large number of the storm track biases are linked to the ocean state and its influence on large scale baroclinicity (Priestley et al., 2021a,b) and a question still remains as to how these historical, mean state errors are propagating through the future simulations. Even though low bias models have been shown to have similar projections to high bias models (Zappa et al., 2013), the pre-existing errors still remain. Understanding

the coupling between large scale historical biases and future projections should therefore be investigated in order to be more clearly understood.



*Code availability.* Code is available at the request of the author. The cyclone tracking and compositing algorithm TRACK is available at the request of Kevin Hodges from *http://www.nerc-essc.ac.uk/~kih/TRACK/Track.html*.

*Data availability.* ERA5 reanalysis is available from the Copernicus Climate Change Service Climate Data Store
(https://cds.climate.copernicus.eu/#!/search?text=ERA5&type=dataset). CMIP6 data is publicly available through the Earth System Grid Federation (https://esgf-node.llnl.gov/projects/cmip6/)

*Author contributions.* MP and JL devised, wrote, and edited the study. Analysis and creation of figures was completed by MP.

*Competing interests.* The authors declare that they have no competing interests.

*Acknowledgements.* M. D. K. Priestley and J. L. Catto are supported by the Natural Environment Research Council (NERC) grant
NE/S004645/1.





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





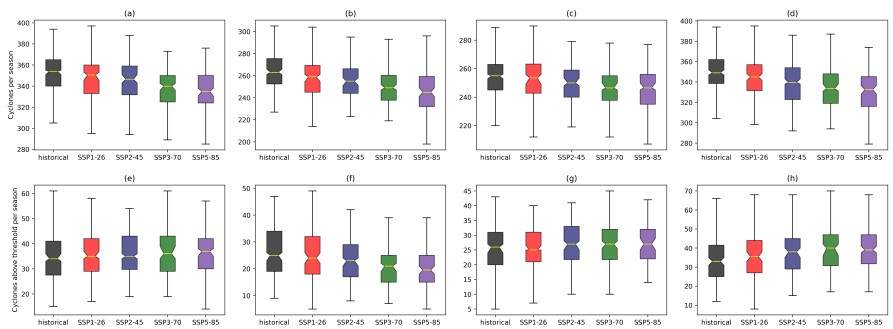

**Figure 1.** Boxplots of (a–d) the total number of identified extratropical cyclones in the CMIP6 models and (e–h) those that exceed the historical 90th percentile of peak cyclone T42 vorticity for the historical (black), SSP1-26 (red), SSP2-45(blue), SSP3-70 (green), and SSP5-85 (purple) experiments. Boxplots are shown for NH DJF (a,e), NH JJA (b,f), SH DJF (c,g), and SH JJA (d,h). The evaluated periods are 1979-2014 for the historical simulation and 2080-2099 for the different SSPs. The yellow lines in the boxes are the median, and the boxes extend to the 25th/75th percentiles. Whiskers extend to the 1.5 times the inter-quartile range (IQR). Notches on the boxes represent the 5-95% confidence range on the median, based on 10,000 bootstrap re-samples. Units for all boxes and all panels are cyclones per season.




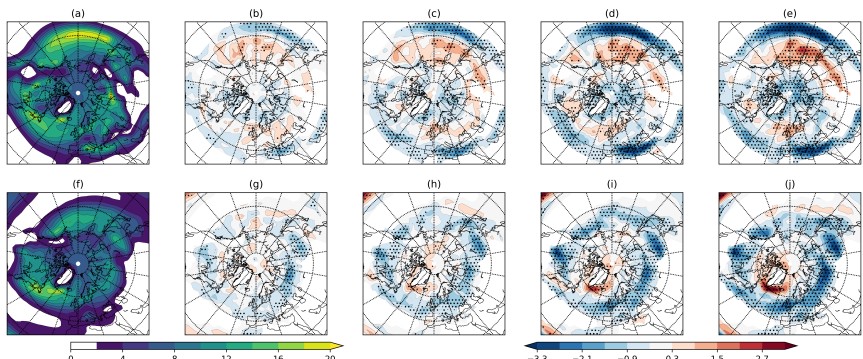

**Figure 2.** Storm track densities for the Northern Hemisphere for DJF (a–e) and JJA (f–j). Seasonal storm track densities are shown for the CMIP6 historical multi-model mean from 1979-2014 (a,f). End of century (2080-2100) changes relative to the historical mean are shown for the SSP1-26 (b,g), SSP2-45 (c,h), SSP3-70 (d,i), and SSP5-85 (e,j) scenarios. Stippling on figures indicates where 80% of the models agree on the sign of the change relative to the historical multi-model mean. Units are number of cyclones per 5° spherical cap per month.





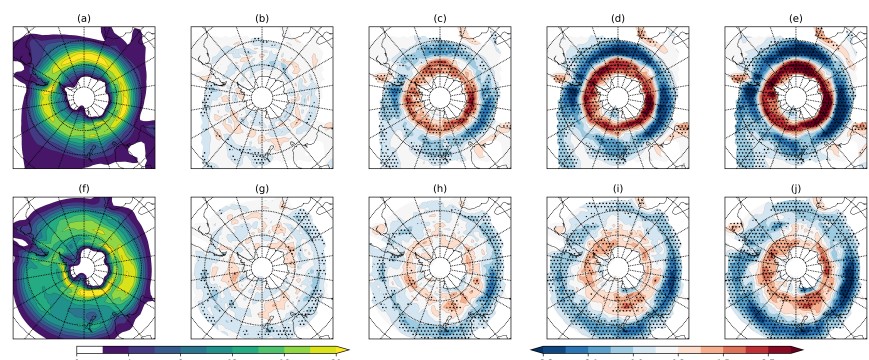

**Figure 3.** As Figure 2 but for the Southern Hemisphere.

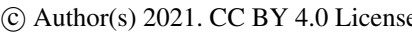



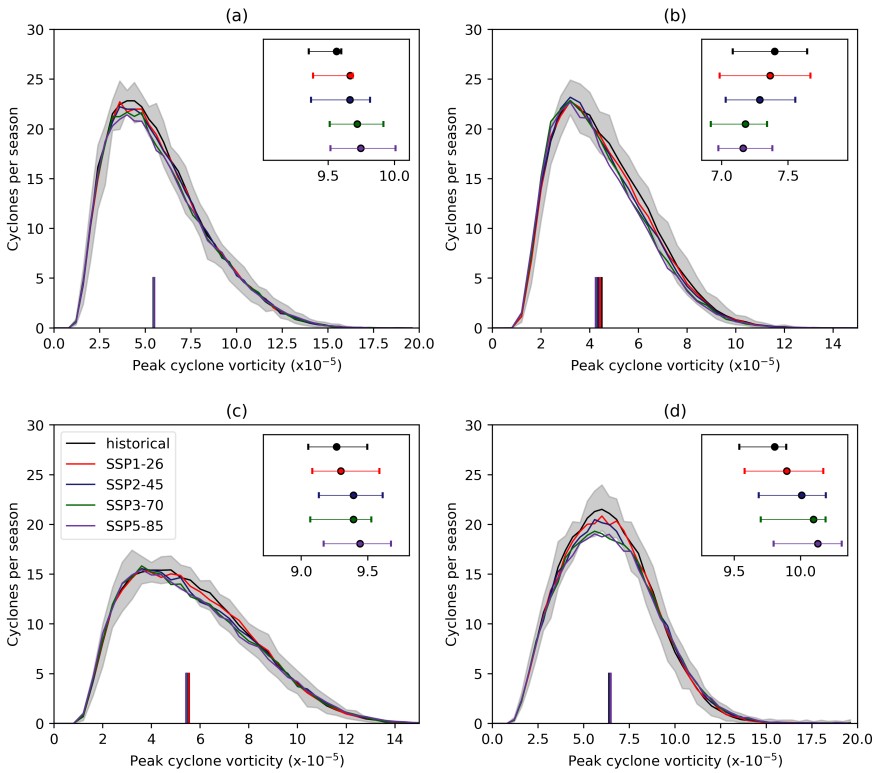

**Figure 4.** Distributions of peak cyclone T42 relative vorticity for (a) NH DJF, (b) NH JJA, (c) SH DJF, and (d) SH JJA. The gray shaded region represents the 5th–95th percentile of the CMIP6 historical models with the black line being the multimodel mean. Coloured lines illustrate the 2080-2100 distribution for SSP1-26 (red), SSP2-45(blue), SSP3-70 (green) and SSP5-85 (purple). The coloured columns from the x axis indicate the median of the distribution of each of the model experiments. The inset for each figure illustrates the 90th percentile of each experiment distribution and associated 25th-75th percentile uncertainty on this value. Units are $s^{-1}$.



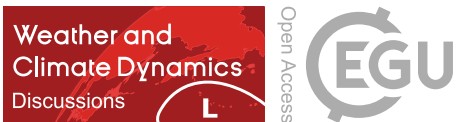

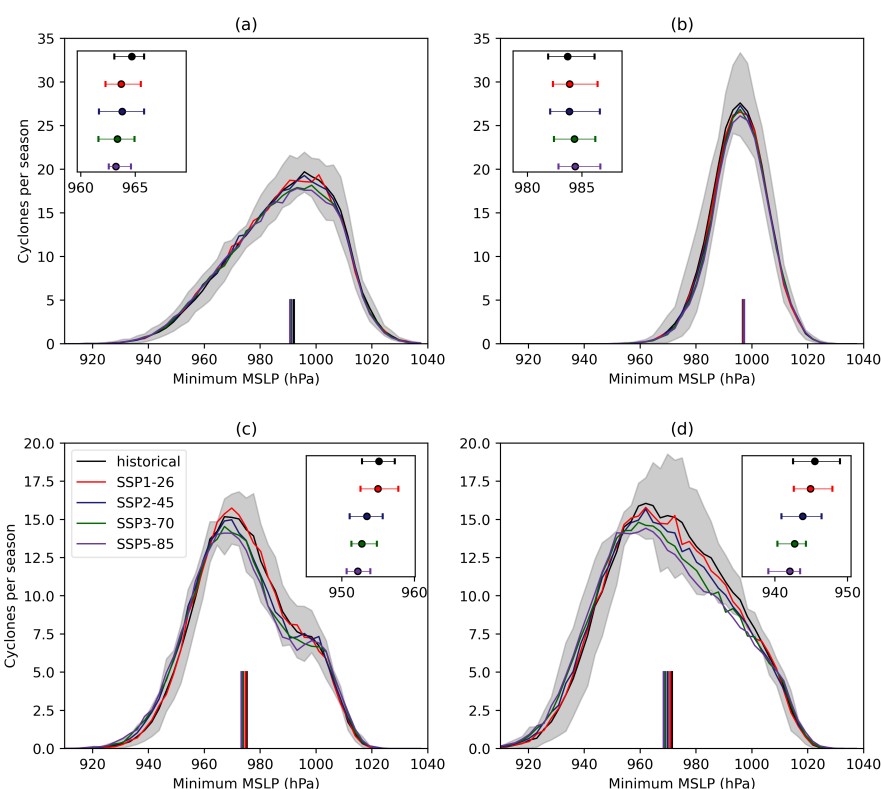

**Figure 5.** As Fig. 4 but for cyclone minimum MSLP (hPa).

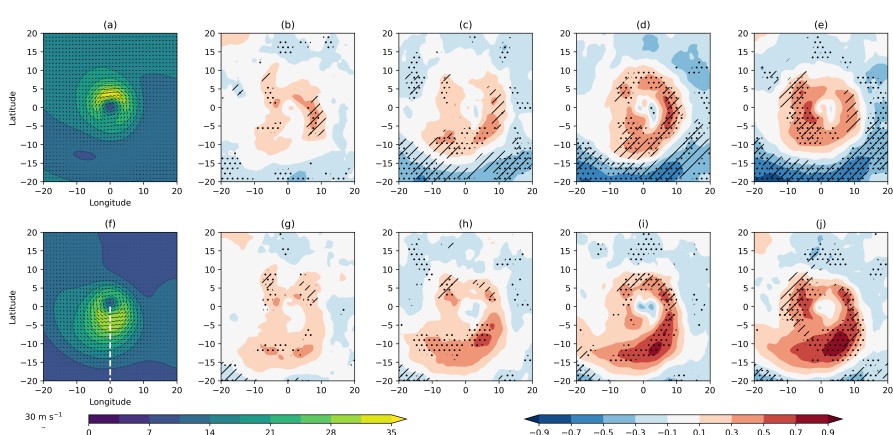

**Figure 6.** Composites of wind speeds at 850 hPa associated with extreme cyclones in a system relative (a–e) and Earth relative (f–j) perspective for NH DJF. Composites are shown for the CMIP6 historical average (a,f) and biases for the SSP1-26 (b,g), SSP2-45 (c,h), SSP3-70 (d,i), and SSP5-85 (e,j) scenarios for the 2080-2100 period. Stippling indicates 80% model consensus on the sign of the bias and hatching shows where the model mean is larger than the model variance. Units are m s$^{-1}$. Dashed white line in (f) is the location of the transect taken in Fig. 10.





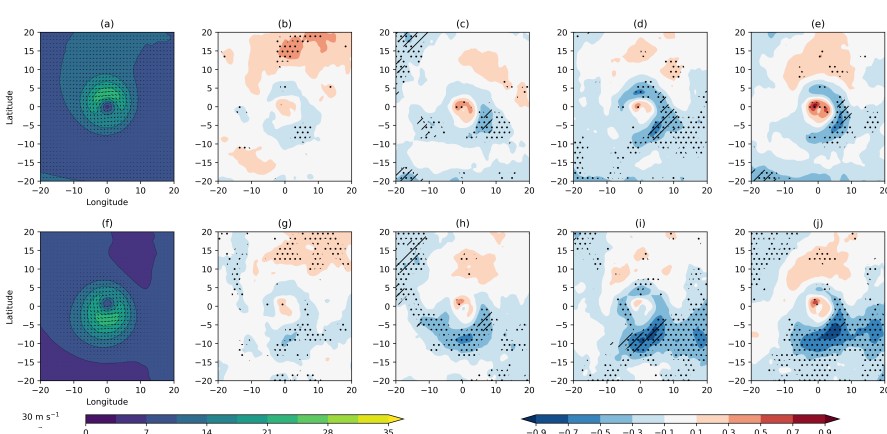

**Figure 7.** As Figure 6 but for NH JJA.





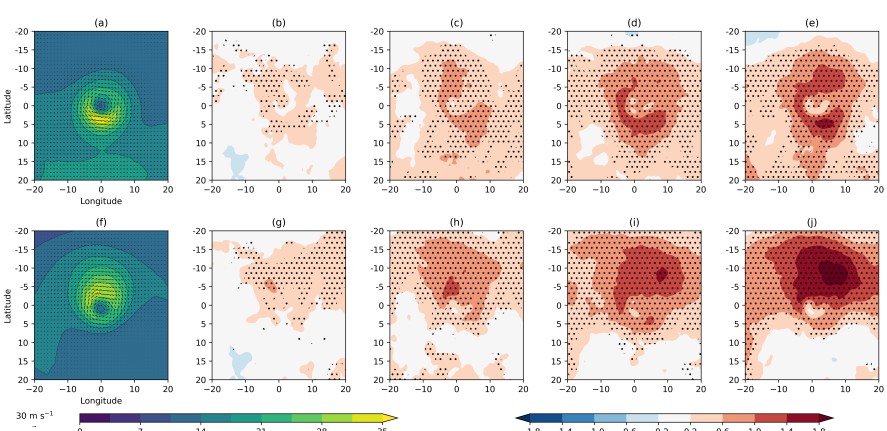

**Figure 8.** As Figure 6 but for SH JJA.



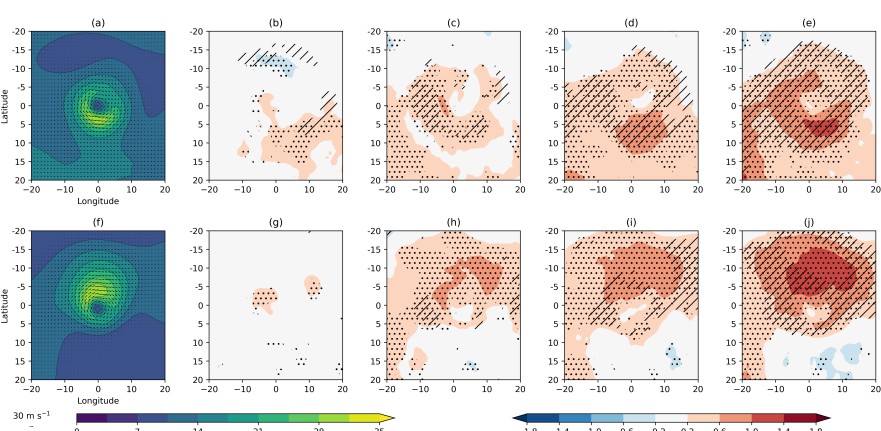

**Figure 9.** As Figure 6 but for SH DJF.



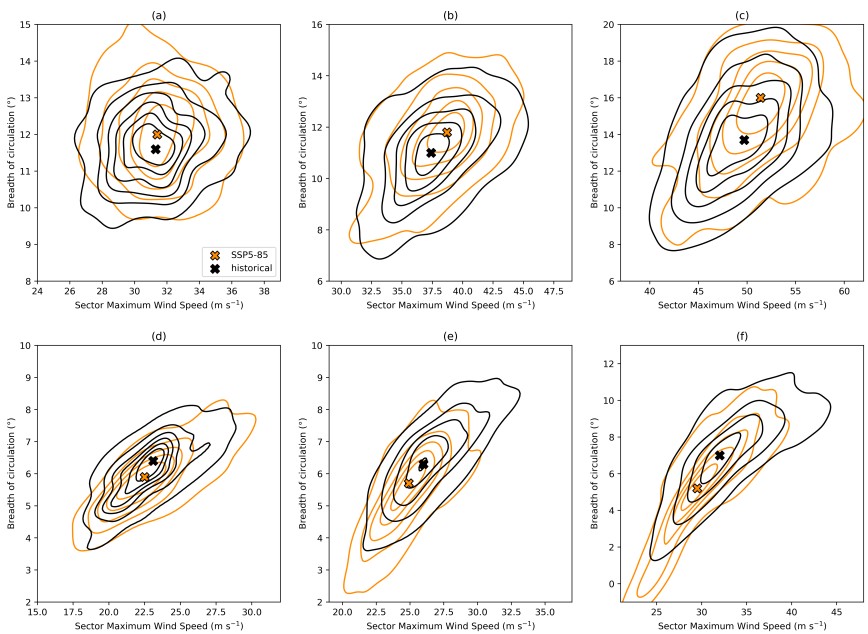

**Figure 10.** Scatter plot of maximum wind speed along the cyclone transect (dashed white line Fig. 6f) against the broadness of wind speeds above a fixed threshold along the same transect plotted via a gaussian kernel density estimator (KDE). KDEs are plotted for the historical (black, 1979-2014) and SSP5-85 experiments (orange, 2080-2100) for the NH in (a–c) DJF and (d–f) JJA for (a,d) 850 hPa, (b,e) 500 hPa, and (c,f) 250 hPa. Crosses indicate the highest frequency of the KDE. Thresholds for the broadness of the circulation are (a,d) 17 m s$^{-1}$, (b) 25 m s$^{-1}$, (c) 35 m s$^{-1}$, (e) 20 m s$^{-1}$ and (f) 25 m s$^{-1}$.



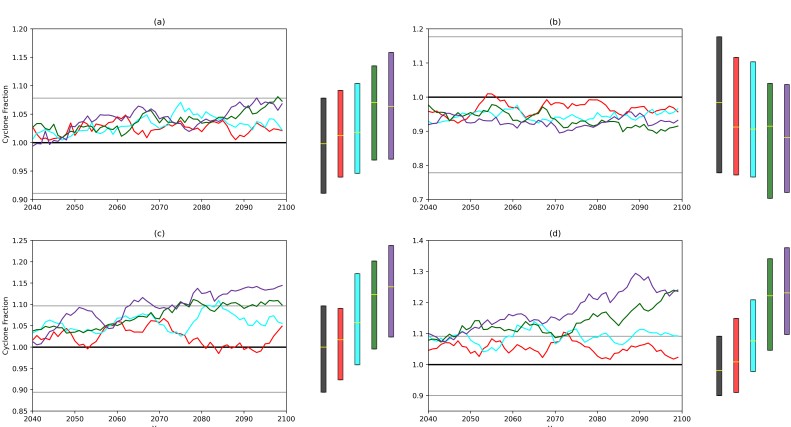

**Figure 11.** Evolution of cyclone area with Earth relative wind speeds at 850 hPa exceeding 17 m s$^{-1}$ from 2040-2100 for SSP1-26 (red), SSP2-45 (cyan), SSP3-70 (green), and SSP5-95 (purple). Timeseries are the multi-model means and are shown for NH DJF (a), NH JJA (b), SH DJF (c), and SH JJA (d). The thick black line refers to the 1979-2014 seasonal average from the historical simulation, with the grey lines being the 25th and 75th percentiles respectively. A cyclone area of 1 is the average area for the 1979-2014 period. Scenario timeseries have been smoothed using a 5-year moving average. Boxplots show the distribution of yearly averaged cyclones areas for each experiment from 2090-2100 (scenarios) and 1979-2014 (historical). The yellow lines indicating the median and boxes extend to the inter-quartile range.





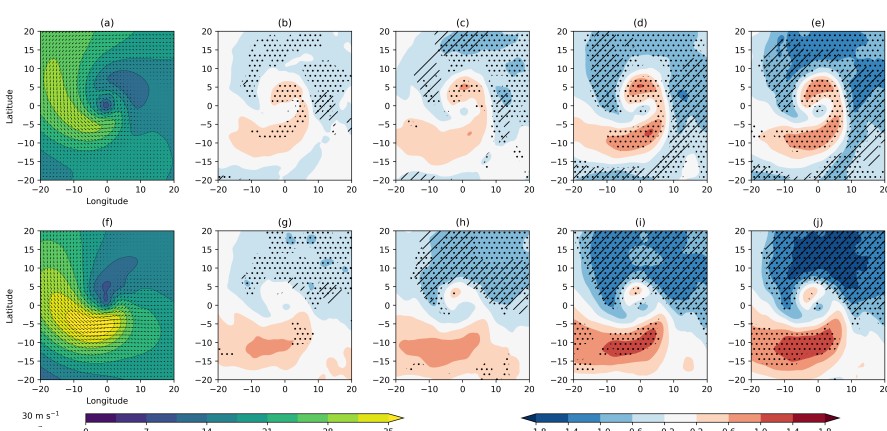

**Figure 12.** As Figure 6 but for 500 hPa.




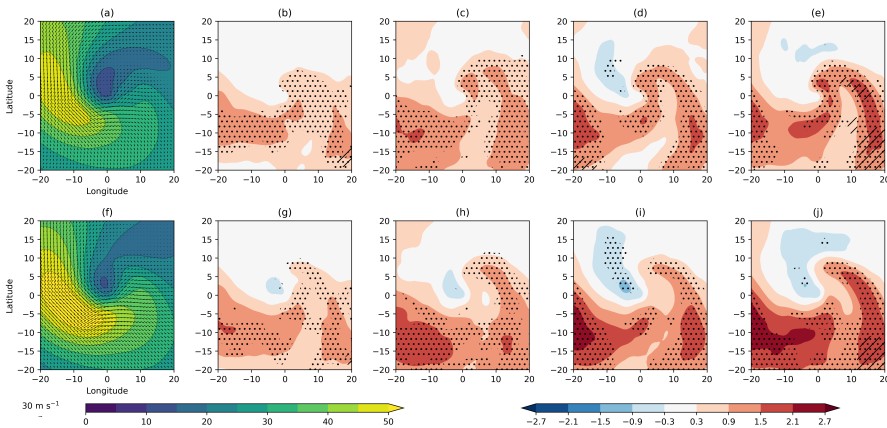

**Figure 13.** As Figure 6 but for 250 hPa.



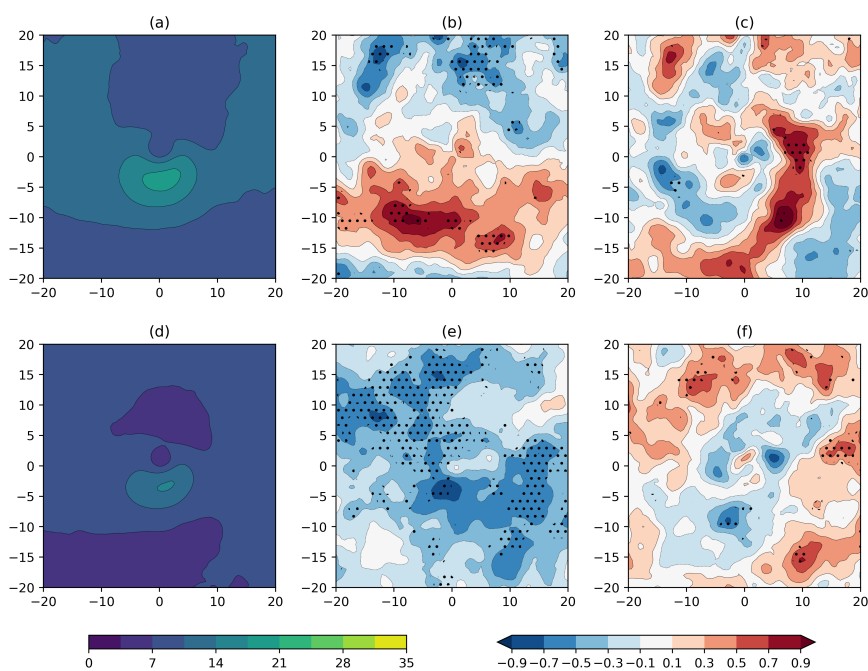

**Figure 14.** Comparison of composite Earth relative wind speed for cyclones over the North Atlantic (30°N-70°N, 280°E-360°E) for (a–c) DJF and (d–f) JJA. (a,d) Composites of the AVG cyclones in the 1979-2014 historical period in ERA5. (b,e) Change in the AVG cyclones in the SSP5-85 scenario for the 2080-2100 period. (c,f) Difference between the changes in the EXT and AVG cyclones for the 2080-2099 period. Stippling indicates where 80% of the models agree on the sign of the change. Units are m s$^{-1}$.



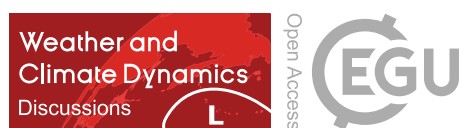

**Table 1.** Table of models used that contribute data to the *historical, SSP1-26, SSP2-45, SSP3-70,* and *SSP5-85* experiments. Gridpoint resolutions stated are that of the gaussian grid the *u* and *v* data are provided on. The quoted nominal resolution in kilometres is the stated nominal resolution of the atmospheric component of the model from Taylor et al. (2017).

| Model Name | Centre | Gridpoints (lon x lat) | Nominal Atmospheric Resolution |
|---|---|---|---|
| ACCESS-CM2 | CSIRO-ARCCSS | 192 x 145 | 250 km |
| BCC-CSM2-MR | BCC | 320 x 160 | 100 km |
| EC-Earth3 | EC-Earth-Consortium | 512 x 256 | 100 km |
| MIROC6 | MIROC | 256 x 128 | 250 km |
| MPI-ESM1-2-HR | MPI-M, DWD, DKRZ | 384 x 192 | 100 km |
| MPI-ESM1-2-LR | MPI-M, AWI, DKRZ | 192 x 96 | 250 km |
| MRI-ESM2-0 | MRI | 320 x 160 | 100 km |
| NorESM2-LM | NCC | 144 x 96 | 250 km |
| NorESM2-MM | NCC | 288 x 192 | 100 km |