# Peer review of "Future changes in the extratropical storm tracks and cyclone intensity, wind speed, and structure"

_Weather and Climate Dynamics, 2021_

## Referee Comment (RC1)

*Review of manuscript 2021-75 submitted to Weather and Climate Dynamics*

**Future changes in the extratropical storm tracks and cyclone intensity, wind speed, and structure**

by Priestley M. and Catto J.

**General comments:**

This manuscript presents an extended analysis of the changes in the intensity of extratropical cyclones applying different metrics in a warming climate using the CMIP6 models. Low, middle and upper wind speeds associated with the extreme cyclones are also investigated.

The highlight of the present study is the vast number of climate models and scenarios. The results confirm with high robustness what previous studies have been hypothesized using idealized models. The frequency of extreme cyclones will increase (except summer in NH) and produce a broad footprint with stronger winds near the surface over the warm region for most of the seasons and hemispheres.

I find the present manuscript well-written, the analysis consistent and the topic of interest to the community of WCD. Therefore, the manuscript should be published. Minor issues have been found that need to be addressed before the manuscript can be published. My detailed comments are found below.

**Specific comments:**

1) You mentioned that changes in the meridional component play an essential role in the wind response at the upper level. Still, it is not possible to identify it in your figures. Would it be possible to add arrows to your wind anomaly composites? or maybe show changes in the v-component?

2) Differences have been found between the system relative and earth relative composites, which means that the storms will travel faster in NH DJF. Do you know if the increase in the storm's speed is the same at the three different levels? I wonder if the cyclone will tend to tilt as the climate warms.

3) Why will the speed of the cyclone in summer decrease while in winter it will increase in the NH? Can you explain a bit more?

4) Line 283 "*The reduction in wind speeds surrounding the core of the cyclone are likely a result of the reduced cyclone pressure gradient (not shown).*" - A reduction in wind speed is related to a weaker pressure gradient, just by balance, but why will the cyclone pressure gradient be weaker? Do you have an explanation?

5) "*Figure 11 shows timeseries of the winter and summer seasons in the NH and SH for the area of the cyclone composite above a fixed threshold of 17 m/s*". - Why did you choose the threshold of 17 m/s?

**Technical corrections:**

- Can you define SSP and CMIP6? It is easier to read for people who are not familiar with these terms.

- Line 283 "*The reduction in wind speeds surrounding the core of the cyclone are likely a result of the reduced cyclone pressure gradient (not shown)."* Can you show it? is it possible to plot SLP or geopotential height? - "is" instead of "are"

- Line 287 "This furtherweakening suggests that (as with the strengthening in the same location in NH DJF) that there are changes in cyclone speed, and therefore a slowing down of cyclones in NH JJA in the future" - "that" appears twice

-Line 503 "are absent or smaller in magnitude that the EXT cyclones" - "than" instead of "that"

-Line 497 "*… from the cyclone centre (Fig. 13c,f)…*" - Fig. 14c,f ?

-Supplement: *Figure S3. Composites of cyclone wind speeds at 850 hPa…* - is it at 500 hPa?

---

## Referee Comment (RC2)

**Review- Future changes in the extratropical storm tracks and cyclone intensity, wind speed, and structure**

I have reviewed the current manuscript by Matthew D. K. Priestley and Jennifer L. Catto. In this study the authors investigate cyclone-centered composites in an ensemble of 9 CMIP6 simulations, with 4 different climate change scenarios. They investigate changes in cyclone numbers and intensity using different measures, such as changes in maximum vorticity and wind speeds, in summer and winter seasons in both hemispheres. Overall, this is a well-written paper that presents a thorough analysis of future cyclone-related changes. While the results mainly recover previous results from CMIP5 models, it is still important to investigate these changes in the updated CMIP6 data. I therefore think the paper is suitable for publication, after some minor revision.

Comments:

- You suggest that the decrease in the number of cyclones can be understood through the reduced lower tropospheric baroclinicity that is a result of Polar Amplification with global warming. However, if this is true, then why do the number of extreme cyclones increases? If baroclinicity is reduced I would expect the number of extreme cyclones to decrease as well. Also, why is the summer response in the NH different? Polar Amplification also occurs in summer in the NH.

- Lines 201-202: "Furthermore, the amplification of polar temperature in the NH is not projected to be as large in the SH (Fan et al., 2020), further maintaining the mid-latitude baroclinicity." - I don't understand this sentence. The polar amplification is largest in the NH, not in the SH.

- For the intensity calculations that are based on vorticity, you can consider taking the total vorticity (and not the deviation from the large scale flow), to see if results are robust (perhaps some of the changes are associated with changes in the large scale flow?).

- Lines 275-276: "This increased speed, coupled with a strengthening of system wind speeds, may lead to increased wind impacts." Actually, the impact can be smaller if they move faster- it means that the duration or persistence of the storms decreases.

- Your claims about the increase/decrease in propagation speeds- you can calculate these directly from the statistics part of the tracking algorithm, for the mean speed of the cyclones.

- The suggested slowing of the cyclones in JJA is consistent with K. Kornhuber and T. Tamarin-Brodsky 2021 ("Future Changes in Northern Hemisphere Summer Weather Persistence Linked to Projected Arctic Warming") and other studies suggesting a slow-down of the midlatitude circulation during NH summer, and could be a result of the poleward shift of the large-scale jet.

- The suggested argument about the strengthening of upward vertical velocities and weakening of downward motions is consistent with the findings of O'Gorman et al. 2018 ("Increase in the skewness of extratropical vertical velocities with climate warming: Fully nonlinear simulations versus moist baroclinic instability") for idealized GCMs, and with T. Tamarin-Brodsky and O. Hadas 2019 ("The Asymmetry of Vertical Velocity in Current and Future Climate"), for cyclones and anticyclones in a comprehensive CMIP5 model. It is related to the increase in dry static stability (influencing the downward motions) vs. the decrease in the "reduced" or "effective" stability which decreases locally in updraft regions due to the effect of moisture.

- Line 460: "...and anticyclonic turning near/at the tropopause."- where do you see this in the figure?

- The section about the North Atlantic cyclones- I think you should compare, wherever possible, your results to Dolores-Tesillos et a. 2021 ("Future changes in North Atlantic winter cyclones in CESM-LENS. Part I: cyclone intensity, PV anomalies and horizontal wind speed") who performed a similar analysis in a different set of simulations.

- "A large amount of the change in wind speed in the Earth relative framework appears to be due to changes in the propagation speed of cyclones."– again, you can check this directly from the statistics tool for the mean speeds of cyclones.

- "...and instead variables that are less influenced by the large-scale atmospheric state, such as winds or vorticity, are used."- I'm not sure how true it is that the wind field is a variable which is less influenced by the large-scale atmospheric state..

- Figures in general: wherever possible, please make fonts and other marks larger.

- Figure 11 caption: SSP5-95=SSP5-85.

---

## Author Response (AR1)

**Reviewer response for manuscript WCD-2021-75 'Future changes in the extratropical storm tracks and cyclone intensity, wind speed, and structure' by Matthew Priestley and Jennifer Catto**

**Reviewer 1**

**General comments:**

This manuscript presents an extended analysis of the changes in the intensity of extratropical cyclones applying different metrics in a warming climate using the CMIP6 models. Low, middle and upper wind speeds associated with the extreme cyclones are also investigated.

The highlight of the present study is the vast number of climate models and scenarios. The results confirm with high robustness what previous studies have been hypothesized using idealized models. The frequency of extreme cyclones will increase (except summer in NH) and produce a broad footprint with stronger winds near the surface over the warm region for most of the seasons and hemispheres.

I find the present manuscript well-written, the analysis consistent and the topic of interest to the community of WCD. Therefore, the manuscript should be published. Minor issues have been found that need to be addressed before the manuscript can be published. My detailed comments are found below.

We thank the reviewer for their kind comments regarding our manuscript. We have addressed all the comments below and noted any changes in the manuscript. We appreciate the comments to include calculations on the cyclone propagation speed as this has improved our analysis and provides clearer results.

**Specific comments:**

1) You mentioned that changes in the meridional component play an essential role in the wind response at the upper level. Still, it is not possible to identify it in your figures. Would it be possible to add arrows to your wind anomaly composites? Or maybe show changes in the v-component?

We have added the meridional component of the wind (Fig. R1 below) to the supplement of the paper. In this figure you can see the increased strength of the southerly meridional component of the wind near the cyclone centre in DJF, and vice versa in JJA. There is also the strengthening of the northerly meridional wind in the southeast of the composite region in DJF, with a weakening in JJA. This is where we see stronger winds in the full composite. As this is the region where winds are turning, the stronger meridional component in DJF indicates that this turning is stronger. We have updated our discussion of this in the main text and included the figure in the supplement (Fig. S6).

[Figure]

*Figure R1. Composites of meridional wind speeds (v) at 850 hPa associated with extreme cyclones in (a–e) DJF and (f–j) JJA in the NH. Composites are shown for the CMIP6 historical average (a,f) and biases for the SSP1-26 (b,g), SSP2-45 (c,h), SSP3-70 (d,i), and SSP5-85 (e,j) scenarios for the 2080-2100 period. Stippling indicates 80% model consensus on the sign of the bias and hatching shows where the model mean is larger than the model variance. Units are m s$^{-1}$.*

2) Differences have been found between the system relative and earth relative composites, which means that the storms will travel faster in NH DJF. Do you know if the increase in the storm's speed is the same at the three different levels? I wonder if the cyclone will tend to tilt as the climate warms.

As we only track the cyclone at 850 hPa we are unable to provide any information as to how cyclone tilt will be changing in the future. Our composites at 500 and 250 hPa use the same latitude and longitude points as at 850 hPa for the cyclone centre. We have clarified this in the methods section of the manuscript.

3) Why will the speed of the cyclone in summer decrease while in winter it will increase in the NH? Can you explain a bit more?

In previous studies the speed of cyclones has often been linked to changes in the overall baroclinicity, which would in turn affect vertical wind shear and speed through the thermal wind relationship. Baroclinicity (as measured by Eady growth rate) has a tendency to decrease in the summer and increase in the winter in the NH, whereas in the SH it increases in both seasons (Lehmann et al., 2014). For lower levels of the troposphere, the enhanced high latitude warming in the NH is to have a larger effect on reducing baroclinicity than in the SH (Chang, 2018; O'Gorman, 2010). For our analysis on the propagation speed we have added new analysis to this manuscript. We calculated the propagation speed from the composites (Fig. R2). We also calculated propagation speeds directly from the tracks and found similar results. Based upon the composite calculation of propagation speed we find that cyclone speeds (at the time of peak intensity) increase in the SH for both seasons. Yet in the NH speeds do not change in NH DJF and decrease in NH JJA. For all seasons the changes to propagation speeds are very small and less than 1 m s$^{-1}$ on average. These findings are

consistent with the papers referenced above on baroclinicity changes and our hypotheses in the initial manuscript. However, in NH DJF we do not see the expected increase in propagation speed. This seems to be due to large variations in each model's estimation of the speed change and both increases and decreases being predicted.

[Figure]

*Figure R2. Boxplots of cyclone propagation speed in the direction of motion as calculated from the seasonal mean composites. Boxplots are shown for (a) NH DJF, (b) NH JJA, (c) SH DJF, and (d) SH JJA for the historical (black), SSP1-26 (red), SSP2-45 (blue), SSP3-70 (green), and SSP5-85 (purple) experiments. The evaluated periods are 1979-2014 for the historical simulation and 2080-2099 for the different SSPs. The yellow lines in the boxes are the median, and the boxes extend to the 25th/75th percentiles. Whiskers extend to the 1.5 times the inter-quartile range (IQR). Notches on the boxes represent the 5-95% confidence range on the median from 1000 bootstrap re-samples. Speeds are calculated at the time of maximum cyclone intensity. Units are m s$^{-1}$.*

4) Line 283 "The reduction in wind speeds surrounding the core of the cyclone are likely a result of the reduced cyclone pressure gradient (not shown)." - A reduction in wind speed is related to a weaker pressure gradient, just by balance, but why will the cyclone pressure gradient be weaker? Do you have an explanation?

We previously created composites of mean sea level pressure (MSLP) to examine how the surface cyclone was changing (Fig. R3). It can be seen in JJA in the NH how the centre of the cyclone increases in MSLP in the future, yet the region 5-15° from the cyclone centre has a pressure decrease. Therefore, the winds within 10° of the cyclone centre are in a weaker pressure gradient and this is likely to contribute significantly to the weakening of the near-surface winds. The reason for the changes in pressure at the cyclone centre are likely due to reduced baroclinicity, and less available energy for the cyclones, whereas the decrease away from the centre may be due to changes in the large-scale pattern. In DJF the stronger pressure gradient is likely due to the intensification of the cyclone, which is thought to be related to moist processes in winter (Büeler and Pfahl, 2019), and also a change in the large-scale pattern, possibly associated with a poleward shift. Furthermore, as the change in the large-scale pattern is often dependent on each models background state, it can be hard to get robust results from the MSLP composite. Therefore, due to further questions being posed by the inclusion of this figure we elected to not include it in the final manuscript.

[Figure]

*Figure R3. Composites of mean sea level pressure in the NH for (a—e) DJF and (f—j) JJA. Shown are (a,f) CMIP6 multi-model mean for 1979-2014, and responses for the 2080-2100 period fo (b,g) SSP1-26, (c,h) SSP2-45), (d,i) SSP3-70), and (e,j) SSP5-85). Stippling indicates where 80% of the models agree on the sign of the change and hatching shows where the model mean change is larger than the historical standard deviation. Units are hPa.*

5) "Figure 11 shows timeseries of the winter and summer seasons in the NH and SH for the area of the cyclone composite above a fixed threshold of 17 m/s". - Why did you choose the threshold of 17 m/s?

17 m s$^{-1}$ was chosen as this encapsulates the peak of the composite footprint in NH DJF (Fig. R4). Variable threshold were chosen to reflect the different cyclone intensities in Fig. 10

(initial manuscript). However, for Fig. 11 we believe it was better to use a consistent threshold across all seasons. We have updated in the manuscript how these thresholds were chosen.

[Figure]

*Figure R4. Composite wind speed in NH DJF at 850 hPa. The thick black contour indicates the region of wind speeds above 17 m s⁻¹. Units are m s⁻¹.*

Technical corrections:
- Can you define SSP and CMIP6? It is easier to read for people who are not familiar with these terms.
We have changed the manuscript to now define CMIP6 in the first paragraph of the introduction and SSP in the methods section.

- Line 283 "The reduction in wind speeds surrounding the core of the cyclone are likely a result of the reduced cyclone pressure gradient (not shown)." Can you show it? is it possible to plot SLP or geopotential height? - "is" instead of "are"
In reference to the reviewer's major point number 4 above. We elect to not show the MSLP composites due to the large-scale influence on the composites.

- Line 287 "This further weakening suggests that (as with the strengthening in the same location in NH DJF) that there are changes in cyclone speed, and therefore a slowing down of cyclones in NH JJA in the future" - "that" appears twice
We have removed the second instance of the word 'that'.

-Line 503 "are absent or smaller in magnitude that the EXT cyclones" - "than" instead of "that"
We have changed this sentence to 'than'.

-Line 497 "… from the cyclone centre (Fig. 13c,f)…" - Fig. 14c,f ?
The figure reference has been corrected and updated to Fig. 14(c,f).

-Supplement: Figure S3. Composites of cyclone wind speeds at 850 hPa… - is it at 500 hPa?
This is correct and has been changed in the supplement.

**References**

Büeler, D. and Pfahl, S., 2019. Potential vorticity diagnostics to quantify effects of latent heating in extratropical cyclones. Part II: application to idealized climate change simulations. Journal of the Atmospheric Sciences, 76(7), pp.1885-1902.

Chang, E. K. M., 2018. CMIP5 projected change in Northern Hemisphere winter cyclones with associated extreme winds. Journal of Climate, 31(16), pp.6527-6542.

Lehmann, J., Coumou, D., Frieler, K., Eliseev, A.V. and Levermann, A., 2014. Future changes in extratropical storm tracks and baroclinicity under climate change. Environmental Research Letters, 9(8), p.084002.

O'Gorman, P.A., 2010. Understanding the varied response of the extratropical storm tracks to climate change. Proceedings of the National Academy of Sciences, 107(45), pp.19176-19180.

**Reviewer 2**

I have reviewed the current manuscript by Matthew D. K. Priestley and Jennifer L. Catto. In this study the authors investigate cyclone-centered composites in an ensemble of 9 CMIP6 simulations, with 4 different climate change scenarios. They investigate changes in cyclone numbers and intensity using different measures, such as changes in maximum vorticity and wind speeds, in summer and winter seasons in both hemispheres. Overall, this is a well-written paper that presents a thorough analysis of future cyclone-related changes. While the results mainly recover previous results from CMIP5 models, it is still important to investigate these changes in the updated CMIP6 data. I therefore think the paper is suitable for publication, after some minor revision.

We thank the reviewer for their comments which have helped us substantially improve this manuscript. We have provided comments to each of their points below with particular focus on new analysis on changes in cyclone intensity and the upper-level divergence. Both these new figures are now present in the revised manuscript and have contributed significantly to our findings.

Comments:

- You suggest that the decrease in the number of cyclones can be understood through the reduced lower tropospheric baroclinicity that is a result of Polar Amplification with global warming. However, if this is true, then why do the number of extreme cyclones increases? If baroclinicity is reduced I would expect the number of extreme cyclones to decrease as well. Also, why is the summer response in the NH different? Polar Amplification also occurs in summer in the NH.

There are decreases in baroclinicity at the surface in both NH winter and summer. Despite this, eddy kinetic energy (EKE) does not decrease in winter due to the increase in upper tropospheric temperature gradient (Lehmann et al., 2014). This difference in overall baroclinicity (minimal change in DJF, decrease in JJA) is part of the reason, however the stronger contributor is likely due to the stronger role of moist processes in the NH winter instead of NH summer. Changes in extremes are often variable dependent of the and both increases and decreases in the number of extreme cyclones have been found (e.g. Chang, 2018; Bengtsson et al., 2009; Zappa et al., 2013; Grieger et al., 2014), which is often dependent on the model used and the area investigated (e.g. Michaelis et al., 2017; Zappa et al., 2013). There have been several idealized studies that have found increases in the strength of the most extreme cyclones (Pfahl et al., 2015; Büeler and Pfahl, 2019; Sinclair et al., 2020). In both these studies it was found that intense cyclones featured stronger lower level potential vorticity (PV) as a result of stronger diabatic processes and an increase in total column water vapour, which likely overrides the decrease in lower-tropospheric baroclinicity. The increase in temperature and therefore moisture content is likely driving this increase in extremes in the future climates. This has also recently been discussed by Dolores-Tesillos et al. (2021) in a coupled GCM, with similar findings of increases of low-level PV in extreme cyclones, driving the changes in intensity. We discuss all these factors in more detail in the revised manuscript in both the results and discussion.

- Lines 201-202: "Furthermore, the amplification of polar temperature in the NH is not projected to be as large in the SH (Fan et al., 2020), further maintaining the mid-latitude

baroclinicity." - I don't understand this sentence. The polar amplification is largest in the NH, not in the SH.

The reviewer is correct. This is a typo in the manuscript and the NH and SH should be reversed. The increase in polar temperature is not as large in the SH, maintaining the baroclinicity more.

- For the intensity calculations that are based on vorticity, you can consider taking the total vorticity (and not the deviation from the large scale flow), to see if results are robust perhaps some of the changes are associated with changes in the large scale flow?).

We use the filtered vorticity for both cyclone identification and tracking. This ensures that all model data is equivalent and we do not get varying values due to the various input resolutions of our model data. Therefore, at no point do we utilise the full vorticity field in any of our analysis. We choose to use the filtered vorticity to eliminate these differences and it ensures that as cyclones move to various latitudes/backgrounds that this does not influence the resultant intensity value. As can be seen in the composite change of MSLP in DJF (Fig R1a—e) there is a strengthening of the pressure gradient across the composite region which is likely associated with the changing latitude of cyclones (this is even more pronounced in the SH). It is likely that a similar artefact would be introduced through analysing the total vorticity. As we have used a number of different measures of intensity, including the filtered vorticity, MSLP, and different wind metrics in the paper, and feel that this gives enough breadth to the analysis without adding the absolute vorticity.

- Lines 275-276: "This increased speed, coupled with a strengthening of system wind speeds, may lead to increased wind impacts." Actually, the impact can be smaller if they move faster- it means that the duration or persistence of the storms decreases.

We agree with the reviewers statement, although this would depend on the balance between the changing propagation speed and the intensity. However, as our new analysis indicates that the cyclones in NH DJF are not changing speed, we have edited this statement to reflect the fact that a consistent speed (relative to historical values) would likely result in higher wind damages.

- Your claims about the increase/decrease in propagation speeds- you can calculate these directly from the statistics part of the tracking algorithm, for the mean speed of the cyclones.

Based on this comment and that of reviewer 1 we evaluated several ways to analyse the cyclone speed. The way that you suggest in using the statistics from the tracking algorithm is one method, however this does not differentiate cyclones at different parts of their lifecycle and would therefore not reflect the state of the cyclones analysed in our composites, which are at the time of maximum intensity. The second is to calculate the speed from the tracks and using the locations 6 hours before and after the time of maximum intensity to calculate an average speed, and the third is to use the composites and calculate the speed via the difference in the earth relative and system relative components. Using the second and thir methods we obtained consistent results and therefore elected to calculate speeds based on the composites to be consistent with the rest of our analysis. We have included this figure in a response to reviewer 1 (Fig. R2) and in the main text of the paper. We find no change to propagation speed in NH DJF, a decrease in NH JJA, and an increase in both seasons in the SH. For NH JJA and in the SH the changes are associated with the changes in large-scale baroclinicity and we discuss this in the manuscript (see also Kornhuber and Tamarin-Brodsky,

2021; O'Gorman, 2010, Chang, 2018). In the NH DJF the picture is less clear as there is a decrease in low-level baroclinicity, but increase at upper-level, with the two likely cancelling each other out (unlike in the SH where the decrease in low-level baroclinicity is markedly weaker). Furthermore, assessment of each models change in speed reveals that for NH DJF the variation is very large, indicating considerable model uncertainty as to the change in propagation speed. This is likely due to the competing factors discussed above. The inclusion of this figure has strengthened the findings of our manuscript and is discussed in detail.

- The suggested slowing of the cyclones in JJA is consistent with K. Kornhuber and T. Tamarin-Brodsky 2021 ("Future Changes in Northern Hemisphere Summer Weather Persistence Linked to Projected Arctic Warming") and other studies suggesting a slow-down of the midlatitude circulation during NH summer, and could be a result of the poleward shift of the large-scale jet.

We that the reviewer for this comment and agree. Our new analysis has indicated that cyclones are slowing down in the NH summer (Fig. R2). We have updated our discussion in the manuscript to reflect this new analysis and to also discuss more how the large-scale baroclinicity is likely impacting the propagation speed. This is discussed in O'Gorman (2010), Chang (2018), Kornhuber and Tamarin-Brodsky (2021).

- The suggested argument about the strengthening of upward vertical velocities and weakening of downward motions is consistent with the findings of O'Gorman et al. 2018 ("Increase in the skewness of extratropical vertical velocities with climate warming: Fully nonlinear simulations versus moist baroclinic instability") for idealized GCMs, and with T.Tamarin-Brodsky and O. Hadas 2019 ("The Asymmetry of Vertical Velocity in Current and Future Climate"), for cyclones and anticyclones in a comprehensive CMIP5 model. It is related to the increase in dry static stability (influencing the downward motions) vs. the decrease in the "reduced" or "effective" stability which decreases locally in updraft regions due to the effect of moisture.

We thank the reviewer for this very helpful comment. Following this we have strengthened our argument on the changes in vertical velocity by analysing the horizontal divergence at 250 hPa (Fig. R5). We see that in NH DJF divergence increases downstream and poleward of the historical maximum, which is consistent with the findings of Sinclair et al. (2020). There are very small changes associated with the maximum convergence, with a slight decrease (increased convergence), which implies only small changes in the strength of the downward motion. These results are therefore consistent with the studies of O'Gorman et al. (2018) and Tamarin-Brodsky and Hadas (2019) and we have chosen to include this figure and the relevant discussion in the revised manuscript. Furthermore, we see a weakening of the divergence outflow of cyclones in NH JJA, indicating that the rate of ascent in these cyclones is weakening. In the SH we see behaviour consistent with NH DJF.

[Figure]

*Figure R5.* Composites of divergence at 250 hPa in the NH for (a—e) DJF and (f—j) JJA. Composites are shown for the (a,f) historical period (1979-2014) and changes in the (b,g) SSP1-26, (c,h) SSP2-45, (d-i) SSP3-70), and (e,j) SSP5-85 experiments for the 2080-2099 period. Stippling indicates where 80% of the models agree on the sign of the change. Units are $s^{-1}$.

- Line 460: "...and anticyclonic turning near/at the tropopause."- where do you see this in the figure?

This comment relates to major comment 1 by reviewer 1. We have included the figure of the meridional component of the wind to highlight that there is a larger negative meridional component of the wind in this SE sector of the composite region, indicating a stronger turning. We have included this figure in the supplement and updated our text in the manuscript.

- The section about the North Atlantic cyclones- I think you should compare, wherever possible, your results to Dolores-Tesillos et a. 2021 ("Future changes in North Atlantic winter cyclones in CESM-LENS. Part I: cyclone intensity, PV anomalies and horizontal wind speed") who performed a similar analysis in a different set of simulations.

We thank the reviewer for pointing out the manuscript by Dolores-Tesillos et al. (2021), which was submitted to the same journal just before our own. We have updated our manuscript in numerous places to acknowledge and discuss their results in the context of our own.

- "A large amount of the change in wind speed in the Earth relative framework appears to be due to changes in the propagation speed of cyclones."– again, you can check this directly from the statistics tool for the mean speeds of cyclones.

We have performed these calculations and included the figure above (Fig. R2) and in the main text. We link this to changes in the large-scale baroclinicity in the discussion of the paper.

- "...and instead variables that are less influenced by the large-scale atmospheric state, such as winds or vorticity, are used."- I'm not sure how true it is that the wind field is a variable which is less influenced by the large-scale atmospheric state..

In this statement we refer to those that may be mis-interpreted by the presence of the background state. For example, MSLP as a measure of intensity is often skewed by the presence of a dominant background pressure gradient. Winds, which may be higher because of the large-scale field, are not artificially higher/lower and are directly related to impacts. We have edited this statement to more accurately communicate this sentence.

- Figures in general: wherever possible, please make fonts and other marks larger.
Figures, fonts, and markers have been made larger throughout the main manuscript and the supplement.

- Figure 11 caption: SSP5-95=SSP5-85.
We have corrected this typo.

**References**

Bengtsson, L., Hodges, K.I. and Keenlyside, N., 2009. Will extratropical storms intensify in a warmer climate?. Journal of Climate, 22(9), pp.2276-2301.

Büeler, D. and Pfahl, S., 2019. Potential vorticity diagnostics to quantify effects of latent heating in extratropical cyclones. Part II: application to idealized climate change simulations. Journal of the Atmospheric Sciences, 76(7), pp.1885-1902.

Chang, E. K. M., 2018. CMIP5 projected change in Northern Hemisphere winter cyclones with associated extreme winds. Journal of Climate, 31(16), pp.6527-6542.

Dolores-Tesillos, E., Teubler, F. and Pfahl, S., 2021. Future changes in North Atlantic winter cyclones in CESM-LENS. Part I: cyclone intensity, PV anomalies and horizontal wind speed. Weather and Climate Dynamics Discussions, pp.1-30.

Grieger, J., Leckebusch, G.C., Donat, M.G., Schuster, M. and Ulbrich, U., 2014. Southern Hemisphere winter cyclone activity under recent and future climate conditions in multi-model AOGCM simulations. International journal of climatology, 34(12), pp.3400-3416.

Kornhuber, K. and Tamarin-Brodsky, T., 2021. Future Changes in Northern Hemisphere Summer Weather Persistence Linked to Projected Arctic Warming. Geophysical Research Letters, 48(4), p.e2020GL091603.

Lehmann, J., Coumou, D., Frieler, K., Eliseev, A.V. and Levermann, A., 2014. Future changes in extratropical storm tracks and baroclinicity under climate change. Environmental Research Letters, 9(8), p.084002.

Michaelis, A.C., Willison, J., Lackmann, G.M. and Robinson, W.A., 2017. Changes in winter North Atlantic extratropical cyclones in high-resolution regional pseudo–global warming simulations. Journal of Climate, 30(17), pp.6905-6925.

O'Gorman, P.A., Merlis, T.M. and Singh, M.S., 2018. Increase in the skewness of extratropical vertical velocities with climate warming: fully nonlinear simulations versus moist baroclinic instability. Quarterly Journal of the Royal Meteorological Society, 144(710), pp.208-217.

O'Gorman, P.A., 2010. Understanding the varied response of the extratropical storm tracks to climate change. Proceedings of the National Academy of Sciences, 107(45), pp.19176-19180.

Pfahl, S., O'Gorman, P.A. and Singh, M.S., 2015. Extratropical cyclones in idealized simulations of changed climates. Journal of Climate, 28(23), pp.9373-9392.

Sinclair, V.A., Rantanen, M., Haapanala, P., Räisänen, J. and Järvinen, H., 2020. The characteristics and structure of extra-tropical cyclones in a warmer climate. Weather and Climate Dynamics, 1(1), pp.1-25.

Tamarin-Brodsky, T. and Hadas, O., 2019. The asymmetry of vertical velocity in current and future climate. Geophysical Research Letters, 46(1), pp.374-382.

Zappa, G., Shaffrey, L.C., Hodges, K.I., Sansom, P.G. and Stephenson, D.B., 2013. A multimodel assessment of future projections of North Atlantic and European extratropical cyclones in the CMIP5 climate models. Journal of Climate, 26(16), pp.5846-5862.